# DynamicVerse: A Physically-Aware Multimodal Framework for 4D World Modeling

**Kairun Wen**[1*†], **Yuzhi Huang**[1*], **Runyu Chen**[1], **Hui Zheng**[1], **Yunlong Lin**[1], **Panwang Pan**[1],
**Chenxin Li**[2], **Wenyan Cong**[3], **Jian Zhang**[1], **Junbin Lu**[4], **Chenguo Lin**[5], **Dilin Wang**[6], **Zhicheng Yan**[6],
**Hongyu Xu**[6], **Justin Theiss**[6], **Yue Huang**[1], **Xinghao Ding**[1✉], **Rakesh Ranjan**[6], **Zhiwen Fan**[3]

\* Equal Contribution; [†] Project Leader; [✉] Corresponding Author
[1]XMU  [2]CUHK  [3]UT Austin  [4]UW  [5]PKU  [6]Meta
**Project Website**: https://dynamic-verse.github.io/

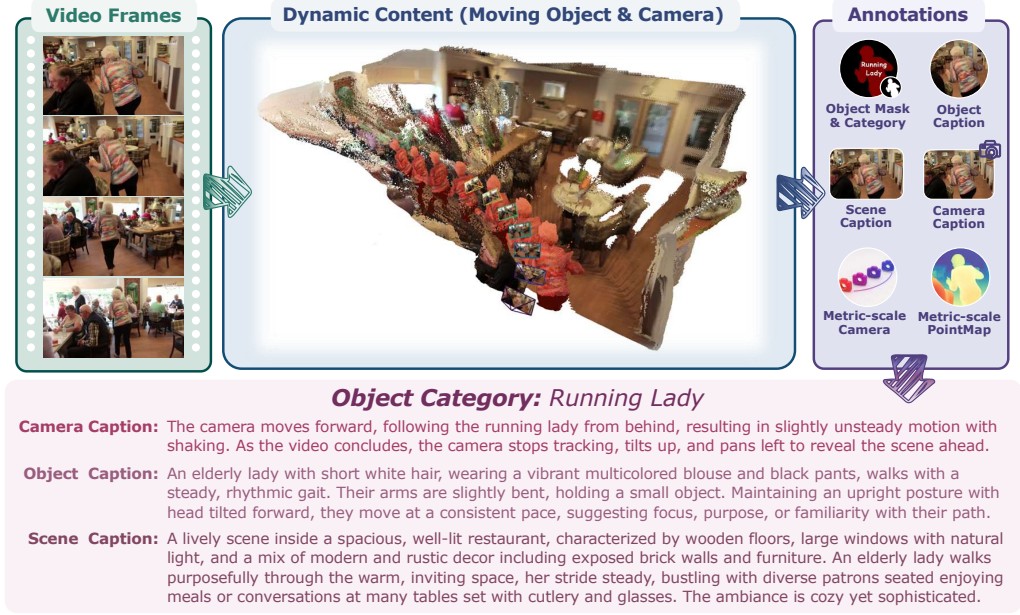

Figure 1: The overview of physically-aware multi-modal world modeling framework **DynamicVerse**.

## Abstract

Understanding the dynamic physical world, characterized by its evolving 3D structure, real-world motion, and semantic content with textual descriptions, is crucial for human-agent interaction and enables embodied agents to perceive and act within real environments with human-like capabilities. However, existing datasets are often derived from limited simulators or utilize traditional Structure-from-Motion for up-to-scale annotation and offer limited descriptive captioning, which restricts the capacity of foundation models to accurately interpret real-world dynamics from monocular videos, commonly sourced from the internet.

To bridge these gaps, we introduce **DynamicVerse**, a physical-scale, multimodal 4D world modeling framework for dynamic real-world video. We employ large vision, geometric, and multimodal models to interpret metric-scale static geometry, real-world dynamic motion, instance-level masks, and holistic descriptive captions. By integrating window-based Bundle Adjustment with global optimization, our method converts long real-world video sequences into a comprehensive 4D multimodal format. DynamicVerse delivers a large-scale dataset consists of 100K+

videos with 800K+ annotated masks and 10M+ frames from internet videos. Experimental evaluations on three benchmark tasks, namely video depth estimation, camera pose estimation, and camera intrinsics estimation, demonstrate that our 4D modeling achieves superior performance in capturing physical-scale measurements with greater global accuracy than existing methods.

# 1 Introduction

Humans inhabit a dynamic 3D world where geometric structure and semantic content evolve over time, constituting a 4D reality (spatial with temporal dimension). Understanding this dynamic environment is fundamental for developing advanced AI applications in fields such as robotics [1, 2, 3, 4, 5, 6], extended reality [7, 8, 9], and digital twins [10, 11]. However, building generalizable foundation models for these downstream tasks faces a longstanding challenge: acquiring high-quality, ground-truth 4D datasets from real-world environments, given that data-driven solutions increasingly demand 4D data while its collection using multiple sensors remains non-scalable. This raises the question: *Can we develop an automated pipeline capable of generating a real-world 4D dataset at scale?*

Current real-world 4D data primarily focus on indoor scenes [12, 13] or autonomous driving scenarios [14], where geometry capture is straightforward, but their diversity is limited. Even synthetic 4D data [15, 16, 17, 18], while controllable, often lack the fidelity and complexity required to truly represent the real world, resulting in a notable simulation-to-real gap. Moreover, physically-aware multimodal annotations—including metric-scale 3D geometry, detailed representations of non-rigid actors (*e.g.*, object size, mask and bounding box, etc.), and descriptive captions of dynamic contents (*i.e.*, object, camera and scene)—are often absent [19, 20]. This limited data landscape, especially when contrasted with the progress fueled by large-scale datasets in modalities like images, videos, and language, underscores *the compelling need for a large-scale, diverse, physically-aware, and semantically rich annotated multi-modal dataset for 4D scene understanding.*

Against this background, this paper aims to generate scalable, physically-aware, and multimodal annotations from massive monocular video data (see Fig. 1) for numerous potential applications, such as enhancing 4D Vision-Language Models [21], facilitating advanced 3D-aware video generation [22], and enabling linguistic 4D Gaussian Splatting [23]. However, achieving this goal is not trivial. To the best of our knowledge, there is currently a significant lack of rich and diverse 4D datasets (see Tab. 1) adequate for these demanding tasks. To address this data scarcity, we introduce **DynamicGen**, a novel automated data curation pipeline (see Fig. 3) designed to generate physically-aware multi-modal 4D data at scale. This pipeline contains two main stages: (1) metric-scale geometric and moving object recovery (*i.e.*, object category and mask) from raw videos, and (2) hierarchical dynamic contents (*i.e.*, object, camera and scene) detailed caption generation. Specifically, the pipeline curates diverse real-world monocular video sources; employs a filtering strategy to remove outliers such as camera motion intensity; integrates multiple foundation models (*i.e.*, VFMs, VLMs, LLMs, GFMs) for initial frame-wise annotation; applies dynamic bundle adjustment to jointly minimize global photometric error; and concludes with dynamic content captioning at three granularities and human-in-the-loop quality review to ensure annotation semantic accuracy.

The resulting multi-modal 4D dataset, termed **DynamicVerse** (see Fig. 1), comprises over 100K distinct 4D scenes, 800K masklets, and 10M video frames. Each scene is extensively annotated with multiple modalities: metric-scale point maps, camera parameters, object masks with corresponding categories, and detailed descriptive captions. We evaluate DynamicGen through three benchmarks: video depth estimation, camera pose estimation, and camera intrinsics estimation. We demonstrate the generalization capability of DynamicGen to process web-scale video data and extract multi-modal information qualitatively. We also conduct human study and GPT-assited evaluation to validate the quality of generated captions.

Our main contributions are summarized as follows:

- We develop **DynamicGen**, a novel automated data curation pipeline designed to generate physically-aware multi-modal 4D data at scale. This pipeline contains two main stages: (1) metric-scale geometric and moving object recovery from raw videos, and (2) hierarchical detailed semantic captions generation at three granularities (*i.e.*, object, camera and scene). Powered by foundation models (*i.e.*, VFMs, VLMs, LLMs, GFMs), DynamicGen efficiently generate 4D data

at scale, thus addressing the critical scalability, physical reality and modality diversity limitations of traditional 4D data curation.

- We introduce **DynamicVerse**, a large-scale 4D dataset featuring diverse dynamic scenes accompanied by rich multi-modal annotations including metric-scale point maps, camera parameters, object masks with corresponding categories, and detailed descriptive captions. DynamicVerse encompasses 100K+ 4D scenes coupled with 800K+ masklets, sourced through a combination of massive 2D video datasets and existing 4D datasets. This represents a significant improvement in terms of data scale, scene and modality diversity compared to prior 4D datasets.

- We validate DynamicGen through three benchmarks: video depth estimation, camera pose and intrinsics estimation. We demonstrate the generalization capability of DynamicGen to process web-scale videos and extract multi-modal information qualitatively. We also conduct human study and GPT-assited evaluation to validate the quality of generated captions.

## 2 Related Work

Table 1: **Comparison of *DynamicVerse* with large-scale 2D video datasets and existing 4D scene datasets.** DynamicVerse expands the data scale and annotation richness compared to prior works.

| Dataset Name | Numerical Statistics | | | Provided Annotations | | | | | | | | Detailed Features | | | |
| | # Videos | # Frames | # Masklets | Camera | Depthmap | Instance Mask | Semantic Mask | Object Category | Object Caption | Scene Caption | Camera Caption | Scene Type | Dynamic Type | Real-world? | Metric-scale? |
|---|---|---|---|---|---|---|---|---|---|---|---|---|---|---|---|
| *2D Video Dataset* | | | | | | | | | | | | | | | |
| DAVIS2017 [24] | 0.2K | 10.7K | 0.4K | ✗ | ✗ | ✓ | ✓ | ✓ | ✗ | ✗ | ✗ | - | - | - | - |
| Youtube-VIS [25] | 3.8K | - | 8,171 | ✗ | ✗ | ✓ | ✓ | ✓ | ✗ | ✗ | ✗ | - | - | - | - |
| UVO-dense [26] | 1.0K | 68.3K | 10.2K | ✗ | ✗ | ✓ | ✗ | ✓ | ✗ | ✗ | ✗ | - | - | - | - |
| VOST [27] | 0.7K | 75.5K | 1.5K | ✗ | ✗ | ✓ | ✗ | ✓ | ✗ | ✗ | ✗ | - | - | - | - |
| BURST [28] | 2.9K | 195.7K | 16.1K | ✗ | ✗ | ✓ | ✓ | ✓ | ✗ | ✗ | ✗ | - | - | - | - |
| MOSE [29] | 2.1K | 638.8K | 5.2K | ✗ | ✗ | ✓ | ✗ | ✓ | ✗ | ✗ | ✗ | - | - | - | - |
| SA-V [30] | 50.9K | 4.2M | 642.6K | ✗ | ✗ | ✓ | ✗ | ✓ | ✗ | ✗ | ✗ | - | - | - | - |
| MiraDATA [31] | 330K | - | - | ✗ | ✗ | ✓ | ✗ | ✓ | ✗ | ✗ | ✗ | - | - | - | - |
| *4D Scene Dataset* | | | | | | | | | | | | | | | |
| T.Air Shibuya [32] | 7 | 0.7K | - | ✓ | ✓ | ✗ | ✓ | ✗ | ✗ | ✗ | ✗ | Mixed | Street | Synthetic | Yes |
| MPI Sintel [33] | 14 | 0.7K | - | ✓ | ✗ | ✗ | ✓ | ✗ | ✗ | ✗ | ✗ | - | Scripted | Synthetic | - |
| FlyingThings3D [34] | 220 | 2K | - | ✓ | ✗ | ✓ | ✓ | ✗ | ✗ | ✗ | ✗ | Mixed | Objects | Synthetic | - |
| Waymo [14] | 1,150 | 200K | - | ✓ | ✓ | ✗ | ✗ | ✗ | ✗ | ✗ | ✗ | Outdoor | Driving | Real-world | Yes |
| CoP3D [12] | 4,200 | 600K | - | ✓ | ✗ | ✗ | ✗ | ✗ | ✗ | ✗ | ✗ | Mixed | Pets | Real-world | - |
| Stereo4D [35] | 110,000 | 10,000K | - | ✓ | ✓ | ✗ | ✗ | ✗ | ✗ | ✗ | ✗ | Mixed | S. fisheye | Real-world | Yes |
| PointOdyssey [15] | 159 | 200K | - | ✓ | ✓ | ✓ | ✗ | ✗ | ✗ | ✗ | ✗ | Mixed | Realistic | Synthetic | Yes |
| Spring [16] | 47 | 6K | - | ✓ | ✓ | ✓ | ✗ | ✗ | ✗ | ✗ | ✗ | Mixed | Realistic | Synthetic | Yes |
| Dynamic Replica [17] | 524 | 145K | - | ✓ | ✓ | ✓ | ✗ | ✗ | ✗ | ✗ | ✗ | Indoor | Realistic | Synthetic | Yes |
| MVS-Synth [18] | 120 | 12K | - | ✓ | ✓ | ✗ | ✗ | ✗ | ✗ | ✗ | ✗ | Outdoor | Urban | Synthetic | Yes |
| RealCam-Vid [19] | 100K | - | - | ✓ | ✓ | ✓ | ✗ | ✗ | ✗ | ✓ | ✗ | Mixed | Realistic | Synthetic | Yes |
| DynPose-100K [20] | 100K | 6,806K | - | ✓ | ✗ | ✗ | ✗ | ✗ | ✗ | ✗ | ✗ | Mixed | Realistic | Synthetic | Yes |
| **DynamicVerse** | 100K+ | 13.6M | 800K+ | ✓ | ✓ | ✓ | ✓ | ✓ | ✓ | ✓ | ✓ | Mixed | Realistic | Real-world | Yes |

**Multi-modal foundation models.** The development of numerous large foundation models in recent years has yielded remarkable performance across multiple tasks such as depth estimation [36, 37, 38, 39, 40], multi-view stereo [41, 42, 43], detection and segmentation [44, 45, 46, 30], human parsing [47], optical flow estimation [48, 49], and point tracking [50, 51, 38]. We propose that these models are highly applicable to achieving holistic 4D understanding, and unifying them within a single framework represents a promising direction for advancing tasks like nonrigid structure from motion. Our DynamicGen pipeline implements this idea by integrating the following pretrained components: UniDepthv2 [52] for geometry initialization, CoTracker3 [51] and UniMatch [49] for correspondence initialization, and Qwen2.5-VL [53] and SA2VA [54] for dynamic object segmentation. This integration, coupled with multi-stage optimization and regularization, allows us to extract accurate metric-scale camera poses and 4D geometry from monocular video. Similar to our method, the concurrently developed Uni4D [55] captures 4D geometry and pose, but it suffers from limited data modalities and discontinuous geometric estimates. In contrast, our *DynamicGen* pipeline not only produces globally refined dense 4D geometry but also supports moving object recovery (*i.e.,* object category and mask) and provides fine-grained dynamic content (*i.e.,* object, camera and scene) caption annotations.

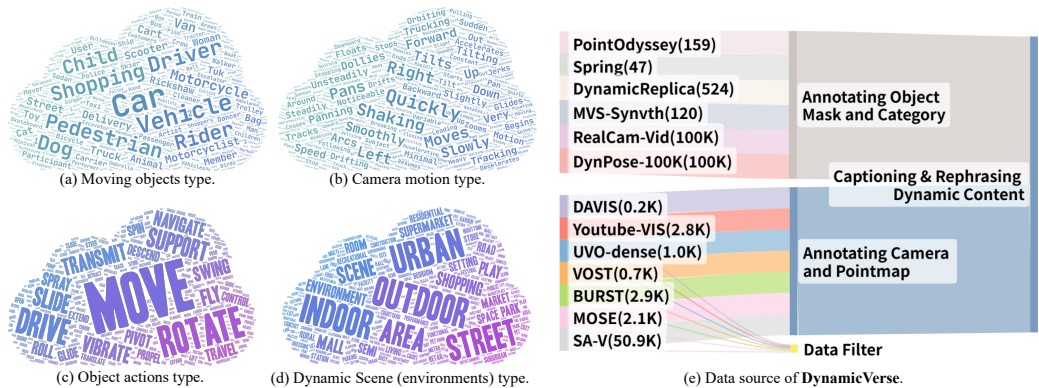

(a) Moving objects type.       (b) Camera motion type.

(c) Object actions type.    (d) Dynamic Scene (environments) type.    (e) Data source of **DynamicVerse**.

Figure 2: The statistics and data source of **DynamicVerse**.

**Multi-modal datasets.** The development of large-scale multi-modal datasets has proven essential for advancing model performance across numerous domains, including language, image-text (*e.g.*, LAION [56, 57], Conceptual Captions [58], WebImageText [59]), and video understanding (*e.g.*, DAVIS2017 [24], Youtube-VIS [25], UVO-dense [26], VOST [27], BURST [28], MOSE [29], SA-V [30], MiraDATA [31]). Extending this success to holistic 4D understanding requires datasets that capture the dynamic 3D world with rich, multi-modal annotations. Existing 4D datasets, whether from early reconstruction efforts [15, 16, 17, 18] (limited diversity) or recent large-scale posed video collections like RealCam-Vid [19] and DynPose-100K [20] (lacking detailed geometry and semantics beyond pose), and even OBJAVERSE [60] (limited content), fall short of providing the comprehensive multi-modal information needed. Our *DynamicVerse* dataset bridges this gap by offering extensive multi-modal annotations, including metric-scale depth, camera parameters, instance segmentation with labels, and descriptive captions, specifically designed to facilitate advanced 4D research.

## 3 DynamicVerse

**Overview** DynamicVerse is a physical-scale, multi-modal 4D modeling framework for real-world video, which contains a novel automated data curation pipeline and corresponding large-scale 4D dataset. The **DynamicGen** pipeline (see Fig. 3) contains two main stages: (1) metric-scale geometric and moving object recovery (*i.e.*, object category and mask) from raw videos, and (2) hierarchical dynamic contents (*i.e.*, object, camera and scene) detailed caption generation. This pipeline primarily consists of five steps: 4D scene curation (in Sec. 3.1), data filter strategy (in Sec. 3.2), moving object recovery (in Sec. 3.3), dynamic bundle adjustment (in Sec. 3.4) and dynamic content caption generation (in Sec. 3.5). The resulting **DynamicVerse** dataset comprises over 100K distinct 4D scenes, 800K masklets, and 10M video frames. The data statistics and collection of DynamicVerse are illustrated in Fig. 2.

### 3.1 4D scene curation

To address the scarcity of available 4D scene data, DynamicGen unifies video data from various real-world video datasets, including DAVIS2017 [24], Youtube-VIS [25], UVO-dense [26], VOST [27], BURST [28], MOSE [29] and SA-V [30], alongside existing synthetic 4D datasets from PointOdyssey [15], Spring [16], Dynamic Replica [17], MVS-Synth [18], RealCam-Vid [19] and DynPose-100K [20]. The inclusion of these datasets is mainly motivated by their potential as scalable data sources for 4D scene understanding.

### 3.2 Data filter strategy

Data filtering is a critical step for identifying video data suitable for subsequent dynamic bundle adjustment. This process presents challenges due to the noisy quality and inherent variability of video data, which impedes the precise selection of high-quality sequences. To address this, we developed a filtering strategy incorporating several distinct criteria: proximal depth, focal-length stability, video blur, camera motion smoothness, and non-perspective distortion. Each of these aspects is quantified

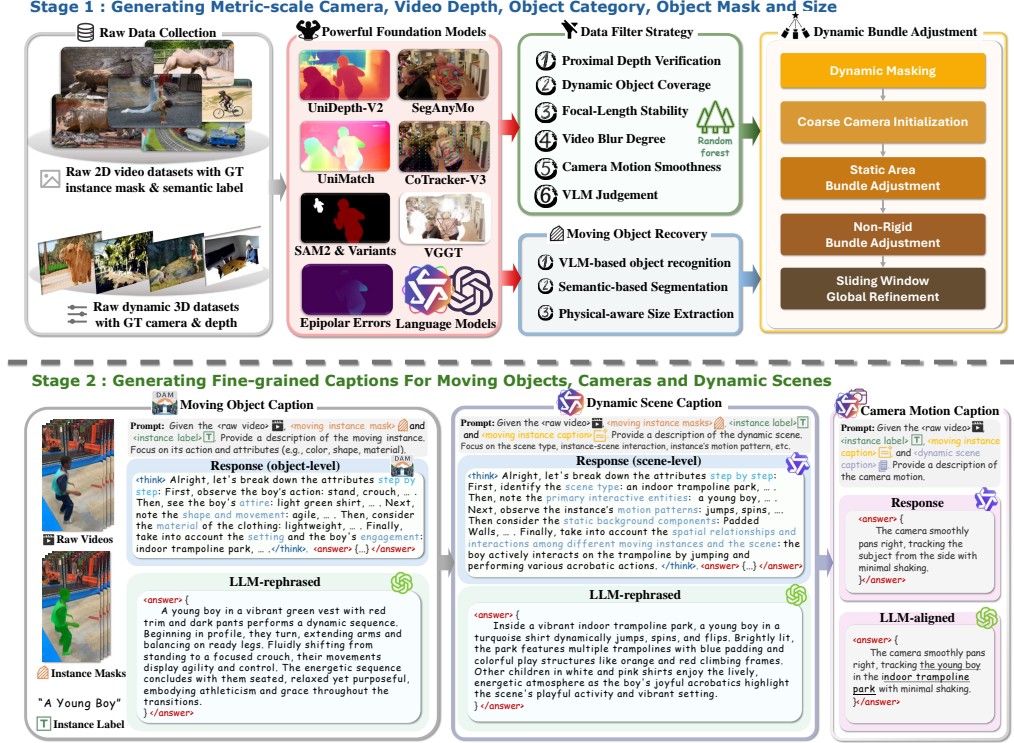

Figure 3: The physically-aware multi-modal 4D data generation pipeline **DynamicGen**.

by a normalized score. We combine these scores as features and employ a Random Forest model to predict a video quality score ranging from 0 to 5. For model training, we manually annotated approximately 1,000 videos, assigning scores between 0 (indicating largely unsuitable, poor quality or insufficient dynamics) and 5 (indicating highly suitable, good quality and sufficient dynamics). We further apply VLM-based judgment to automatically exclude unsuitable videos before reconstruction.

### 3.3 Moving object recovery

To accurately identify the main dynamic objects within a video, we integrated multiple foundation models to achieve reliable segmentation. Specifically, our pipeline first employs Qwen2.5-VL [61] to identify moving objects and determine their semantic categories. These categories are then used to prompt SA2VA [54] for generating corresponding object masks. Leveraging the obtained object masks and geometric annotations, we can apply physical-aware size extraction to annotate the 3D bounding box for moving objects.

### 3.4 Dynamic bundle adjustment

Leveraging the high-quality RGB filtered videos, we employed a robust dynamic bundle adjustment method for annotating metric-scale camera parameters and point maps. This task is challenging due to dynamic objects occluding the static scene and static scene appearance changes hindering correspondence estimation. To effectively addresses both difficulties, we design a multi-stage optimization framework, see Fig. 3, including: (1) dynamic masking, (2) coarse camera initialization, (3) tracking-based static area bundle adjustment, (4) tracking-based non-rigid bundle adjustment, and (5) flow-based sliding window global refinement. Compared with traditional Structure-from-Motion techniques [62] and DUSt3R-based methods [63], our framework not only can handle massive video data with different resolutions but also yield metric-scale results by leveraging the full power of various foundation models.

**Formulation** Given $T$ video RGB frames $\mathbf{I} = (I_1, \ldots, I_T)$ with resolution $H \times W$, we aim to estimate for each timestep $t = 1, \ldots, T$: per-frame pointmap $\mathbf{X}^t \in \mathbb{R}^{H \times W \times 3}$, camera intrinsics $\mathbf{K}^t$, and camera pose $\mathbf{P}^t = [\mathbf{R}^t | \mathbf{T}^t]$, where $\mathbf{R}^t$ and $\mathbf{T}^t$ denote the $t$-th camera's rotation and translation,

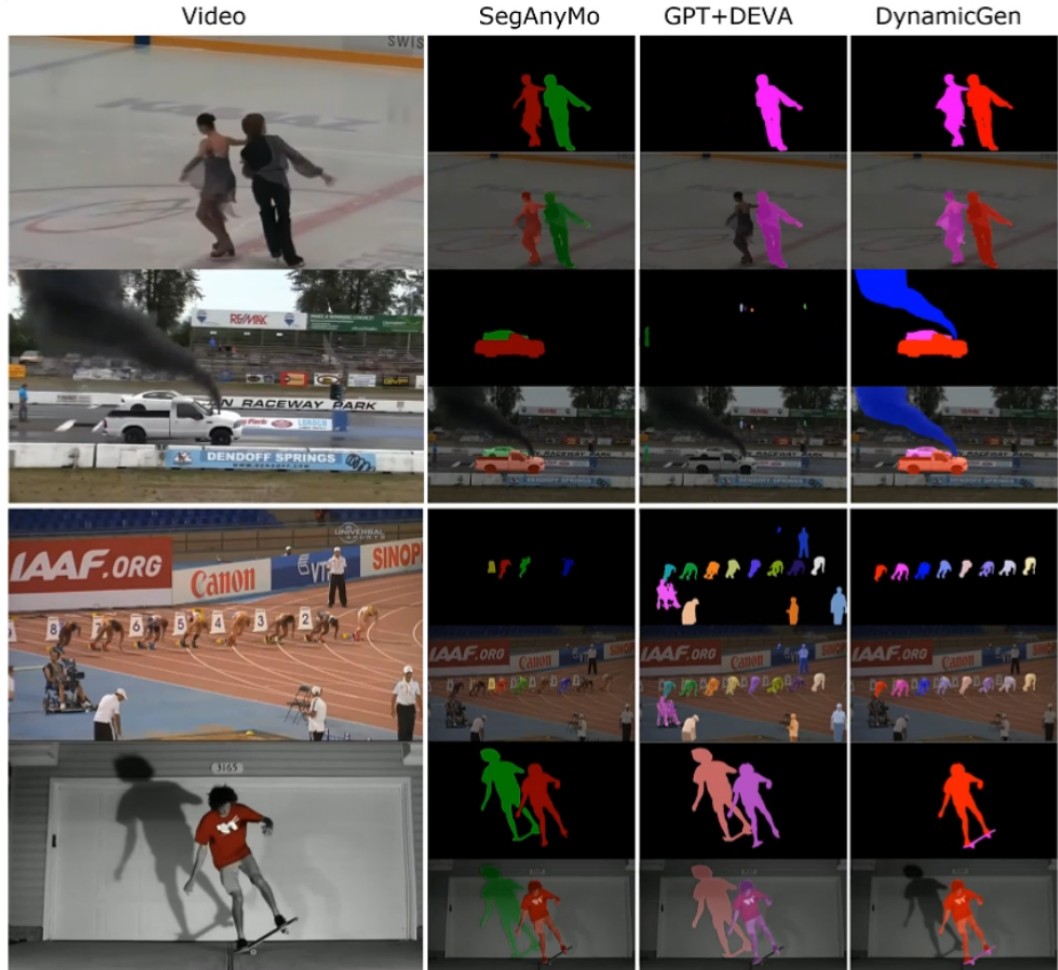

Figure 4: **Qualitative Results of Moving Object Segmentation.** We show qualitatively some of our segmentation results on the Youtube-VIS dataset compared with other methods.

respectively. Here, $\mathbf{X}$ contains static points $\mathbf{X}_{static}$ and dynamic points $\mathbf{X}_{dyn}$. We assume all frames share the same intrinsics $K$ where we optimize focal lengths $f_x$ and $f_y$. The overall cost function is formulated as follows:

$$C_{\text{BA}}(\mathbf{P}, \mathbf{X}_{\text{static}}) + C_{\text{flow}}(\mathbf{X}_{\text{static}}) + C_{\text{NR}}(\mathbf{X}_{\text{dyn}}) + C_{\text{motion}}(\mathbf{X}_{\text{dyn}}) + C_{\text{cam}}(\mathbf{R}) \qquad (1)$$

where $C_{\text{BA}}(\mathbf{P}, \mathbf{X}_{\text{static}})$ and $C_{\text{flow}}(\mathbf{X}_{\text{static}})$ are bundle adjustment terms measuring the reprojection error between static correspondences and the static 3D structure $\mathbf{X}_{\text{static}}$. $C_{\text{NR}}(\mathbf{X}_{\text{dyn}})$ is a non-rigid structure-from-motion term evaluating the consistency of the dynamic point cloud with its tracklets. Regularization is applied to camera motion smoothness through $C_{\text{cam}}(\mathbf{P})$ and to the dynamic structure and motion via $C_{\text{motion}}(\mathbf{X}_{\text{dyn}})$. Each term participates in different optimization stages, which are described below. Detailed explanations of the cost terms are provided in the *supplementary material*.

**Stage I: Dynamic masking**   We first extract dynamic masks to filter out the dynamic points for static area bundle adjustment. Specifically, we use semantic-based and motion-based method to obtain dynamic masks $\mathcal{M} = \{\mathbf{M}^t\}_{t=0}^T = \{\mathbf{M}_{sem}^t \cup \mathbf{M}_{flow}^t\}_{t=0}^T$. For the segmentation-based approach, we use the generated moving object masks $\{\mathbf{M}_{sem}^t\}_{t=0}^T$ in Sec. 3.3. For the flow-based approach, we employ Unimatch [49] to obtain dense optical flow predictions and compute per-frame epipolar error maps [64], which indicate the likelihood of pixels belonging to the dynamic foreground. Then we can obtain dynamic masks $\mathbf{M}_{flow} = [E_1, E_2, \ldots, E_T]$ by thresholding on these epipolar error maps.

**Stage II: Coarse camera initialization** In this stage, we start camera initialization by obtaining video depth $\mathcal{D} = \{\mathbf{D}_t\}_{t=0}^{T}$ and dense pixel motion $\mathcal{Z} = \{\mathbf{Z}_k\}_{k=0}^{K}$. For video depth estimation, we use UniDepthV2 [52], a monocular depth estimation network, to estimate initial depth maps $\mathcal{D}$ and initial camera intrinsics $\mathbf{K}_{\text{init}}$. For dense pixel motion estimation, we utilize Co-TrackerV3 [51] for its robustness. We apply Co-Tracker bi-directionally on a dense grid every 10 frames to ensure thorough coverage. We filter and classify tracklets using segmentation masks yielding a set of correspondent point trajectories $\{\mathbf{Z}_k \in \mathbb{R}^{T \times 2}\}_{k=0}^{K}$ at visible time steps determined by Co-Tracker. Combining $\mathcal{D}$ and $\mathcal{Z}$ allows us to establish 2D-to-3D correspondences. This allows us to initialize and tune camera parameter $\mathbf{P}$ by minimizing the following cost function with respect to camera parameters *only*. Specifically, we can unproject each video frame's depth at time $t$ back to 3D and minimize the following cost function:

$$\min_{\mathbf{P}} \sum_{(t',t)} \sum_{\mathbf{Z}_k \in \neg \mathcal{M}} \|\mathbf{Z}_{k,t'} - \pi_{\mathbf{K}}(\pi_{\mathbf{K}}^{-1}(\mathbf{Z}_{k,t}, \mathbf{D}_t, \boldsymbol{\xi}_t), \boldsymbol{\xi}_{t'})\|_2^2 \tag{1}$$

where $\pi_{\mathbf{K}}^{-1}$ is the unprojection function that maps 2D coordinates into 3D world coordinates using estimated depth $\mathbf{D}_t$. We perform this over all pairs within a temporal sliding window of 5 frames. Given camera initialization $\hat{\mathbf{P}}$, we unproject our depth prediction into a common world coordinate system, which provides an initial 4D structure $\hat{\mathbf{X}}$. This is used as initialization for later optimization.

**Stage III: Static area bundle adjustment** We jointly optimizes camera pose and static geometry by minimizing the static component-related energy in a bundle adjustment fashion. Formally speaking, we solve the following:

$$\min_{\mathbf{P}, \mathbf{X}_{\text{static}}} C_{\text{BA}}(\mathbf{P}, \mathbf{X}_{\text{static}}; \mathcal{Z}, \mathcal{M}) + C_{\text{cam}}(\mathbf{R}) \tag{2}$$

By enforcing consistency with each other, this improves both the static geometry and the camera pose quality. We perform a final scene integration by unprojecting correspondences into 3D using improved pose and filtering outlier noisy points in 3D.

**Stage IV: Non-rigid bundle adjustment** Given the estimated camera pose, this stage focuses on inferring dynamic structure. Note that we freeze camera parameters in this stage, as we find that incorrect geometry and motion evidence often harm camera pose estimation rather than improve it. Additionally, enabling camera pose optimization introduces extra flexibility in this ill-posed problem, harming robustness. Formally speaking, we solve the following:

$$\min_{\mathbf{X}_{\text{dyn}}} C_{\text{NR}}(\mathbf{X}_{\text{dyn}}; \mathbf{P}, \mathcal{Z}, \mathcal{M}) + C_{\text{motion}}(\mathbf{X}_{\text{dyn}}) \tag{3}$$

We initialize $\mathbf{X}_{\text{dyn}}$ using video depth and our optimized camera pose from last step. This energy optimization might still leave some high-energy noisy points, often from incorrect cues, motion boundaries, or occlusions. We filter these outliers based on their energy values in a final step. To further densify the global point cloud, enabling each pixel to correspond to a 3D point, we perform depthbased interpolation by computing a scale offset.

**Stage V: Sliding window global refinement** Given the estimated optical flow, this stage focuses on refining static structure. Note that we freeze camera parameters in this stage. Formally speaking, we solve the following:

$$\min_{\mathbf{X}_{\text{static}}} C_{\text{flow}}(\mathbf{X}_{\text{static}}) \tag{4}$$

With consideration for accuracy and efficiency, the sliding window global refinement is capable of significantly enhancing the multi-view consistency of static points and generalizing effectively to real-world 4D scenes. The detailed process can be found in the *appendix*.

### 3.5 Dynamic Content Caption Generation

Drawing upon the emphasis placed by LEO [65] and SceneVerse [66] on the criticality of caption quality and granularity for comprehensive scene understanding, we design captions at three specific levels: object, scene, and camera. Object captioning focuses on detailed object motion, scene captioning describes object-scene interactions, and camera captioning conveys intricate camera movement. To argument the caption, Large Language Models (LLMs) are employed to automatically rephrase initial captions and align them with these three granularity levels. Finally, to ensure data quality, human verification is conducted to filter out low-quality caption annotations.

**Moving object captioning.** Moving object captions provide detailed descriptions crucial for object grounding. However, prior datasets often have incorrect temporal alignment [66] or insufficient detail [15, 67], while current video captioning methods yield only simple (*e.g.*, Panda-70M [68]) or non-localized descriptions (*e.g.*, Qwen2.5-VL [61]). To address these limitations and generate detailed, accurate captions for individual objects, we utilize DAM [69], known for its superior capabilities. Given RGB videos and corresponding object masks, DAM [69] generates detailed and temporally aligned object descriptions through carefully designed prompts, enabling precise grounding and richer scene understanding.

**Dynamic scene captioning.** Scene-level captions are designed to capture global information, depicting the key objects within the scene along with their associated actions, interactions, and functionalities. For a comprehensive understanding of the entire dynamic scene, we utilize Qwen2.5-VL [61] for dynamic scene captioning. To obtain more detailed, fine-grained, and accurate captions, we propose the use of structured captions. This process involves leveraging the fine-grained moving object captions as auxiliary input and employing specific prompting to generate the final scene-level descriptions. In the design of the prompts, we discovered that an explicit Hierarchical Prompt Design [70] significantly aids the Qwen2.5-VL[61] in comprehending its role, its expected format, and its operational boundaries. This approach contributes to the stabilization of the output's format and enhances the overall quality of the results.

**Camera motion captioning.** Camera Motion Captioning aims to describe the camera's trajectory and movement patterns. Using the powerful VLM [71], we analyze the sequence of inter-frame transformations to identify key motion types like panning, tilting, zooming, and dolly movements. This kinematic information is then used to generate natural language descriptions, potentially leveraging template-based generation or LLM prompting, to convey how the viewpoint changes over time.

**Caption rephrasing.** Following the generation of three distinct caption types (object, scene, and camera motion), a Large Language Model (LLM) [61] is employed to jointly process them. This step aligns the dynamic content descriptions across caption types and refines their phrasing to enhance overall consistency and readability.

**Human-in-the-loop quality review.** To provide a faithful comparison against larger pretrained models, human evaluation was used. Addressing persistent errors from source annotation inaccuracies, we implemented an iterative human-in-the-loop verification during caption construction to identify errors, trace sources, and revise/remove problematic data.

## 4 Experiments

In this section , we present experimental results to evaluate the robustness of our *DynamicGen* pipeline. Due to the page limit, we direct readers to the *appendix* for implementation details, more qualitative results, and more experimental analyses.

### 4.1 Video Depth Estimation

To evaluate video depth estimation accuracy, we assess several baseline methods, including metric depth predictors such as Metric3Dv2 [72], Depth-Pro [36], DepthCrafter [37], and Unidepth [39], which operate without scale or shift alignment. We also consider joint 4D modeling approaches, including MonST3R [63] and RCVD [73]. Evaluations are conducted on the Sintel [33] and KITTI [75] datasets, following standard protocols [37] by applying global shift and scale alignment to the predicted depth maps. We report absolute relative error (Abs Rel) and the percentage of inlier points ($\delta < 1.25$), with all methods undergoing least-squares alignment in disparity space. As shown in Tab. 2, *DynamicGen* achieves the best overall performance across all datasets and evaluation metrics. In particular, it consistently outperforms prior approaches in both absolute accuracy and geometric consistency, demonstrating strong generalization to diverse and dynamic scenes. As illustrated in Fig. 5, MonST3R consistently struggles with object geometry reconstruction, producing distorted

All research undertaken at Meta AI was limited to general guidance on model architectural design. Meta did not participate in any model training activities. Fan, Z. contributed to this project prior to the NeurIPS submission deadline.

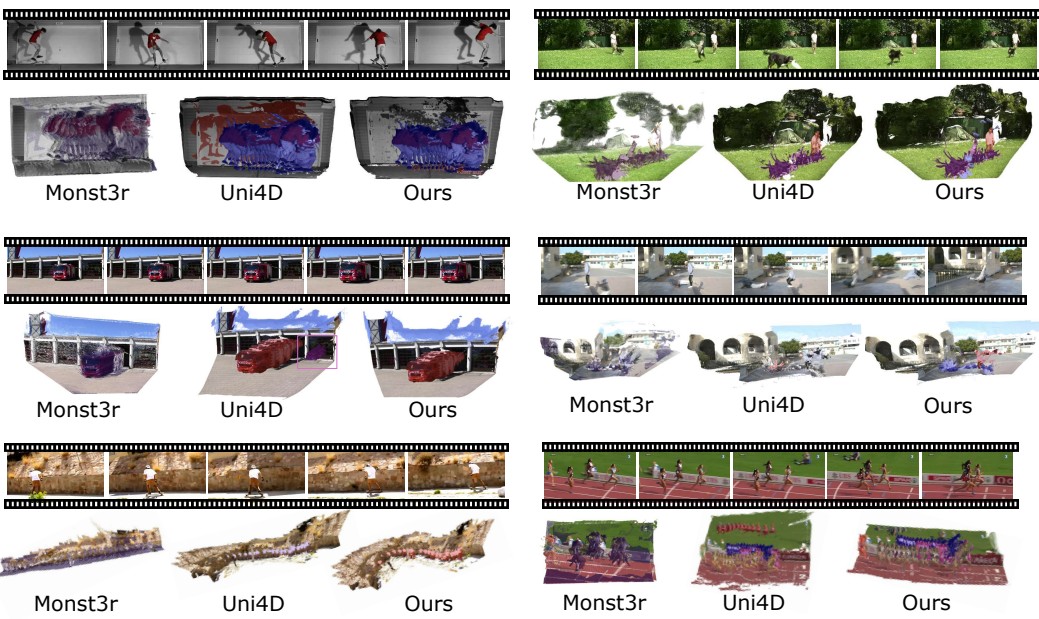

Figure 5: **Visual comparisons of 4D reconstruction on in-the-wild data.**

Table 2: Video depth evaluation on Sintel and KITTI datasets. **Bold** and underlined values indicate best and second best results.

| Alignment | Category | Method | Sintel Abs↓ | Sintel δ1.25↑ | KITTI Abs↓ | KITTI δ1.25↑ |
|---|---|---|---|---|---|---|
| Per-sequence scale | Joint depth & pose | Monst3r [63] | 0.344 | 55.9 | 0.089 | 91.4 |
| | | Uni4D [55] | **0.289** | **64.9** | **0.086** | **93.3** |
| Per-sequence scale & shift | Single-frame depth | Depth-pro [36] | 0.280 | 60.5 | 0.080 | 94.2 |
| | | Metric3D [72] | **0.205** | **71.9** | **0.039** | **98.8** |
| | Video depth | DepthCrafter [37] | 0.231 | 69.0 | 0.112 | 88.4 |
| | Joint video depth & pose | Robust-CVD [73] | 0.358 | 49.7 | 0.182 | 72.9 |
| | | CasualSAM [74] | 0.292 | 56.9 | 0.113 | 88.3 |
| | | Uni4D [55] | 0.216 | 72.5 | 0.098 | 89.7 |
| | | *DynamicGen*(Ours) | **0.205** | **72.9** | **0.091** | **91.2** |

shapes and noisy dynamic masks. Uni4D also exhibits mask imprecision. DynamicGen, however, achieves the cleanest dynamic segmentations and the strongest dynamic/static reconstructions.

## 4.2 Camera Pose Estimation

We evaluate our method against recent dynamic scene pose estimation approaches, including learning-based visual odometry (*e.g.*, LEAP-VO [76], DPVO [77]) and joint depth-pose optimization methods (*e.g.*, Robust-CVD [73], CasualSAM [74], MonST3R [63]). Experiments are conducted on the Sintel [33] and TUM-dynamics [78] datasets, following LEAP-VO's split for Sintel and subsampling the first 270 frames of TUM-dynamics, as done in MonST3R. Camera trajectories are aligned using Umeyama alignment [79], and we report Absolute Trajectory Error (ATE), Relative Translation Error (RPE trans), and Relative Rotation Error (RPE rot). As shown in Tab. 3, *DynamicGen* consistently achieves state-of-the-art results across all metrics and datasets, outperforming existing methods in both translation and rotation accuracy.

Table 3: Camera Pose Evaluation on Sintel and TUM-dynamic datasets. **Bold** and underlined values indicate best and second best results.

| Category | Method | Sintel | | | TUM-dynamics | | |
|---|---|---|---|---|---|---|---|
| | | ATE↓ | RPE trans↓ | RPE rot↓ | ATE↓ | RPE trans↓ | RPE rot↓ |
| Pose only | DPVO [77] | 0.171 | **0.063** | **1.291** | **0.019** | **0.014** | **0.406** |
| | LEAP-VO [76] | **0.035** | 0.065 | 1.669 | 0.025 | 0.031 | 2.843 |
| Joint depth & pose | Robust-CVD [73] | 0.368 | 0.153 | 3.462 | 0.096 | 0.027 | 2.590 |
| | CasualSAM [74] | 0.137 | 0.039 | 0.630 | 0.036 | 0.018 | 0.745 |
| | Monst3r [63] | **0.108** | 0.043 | 0.729 | 0.108 | 0.022 | 1.371 |
| | Uni4D [55] | 0.110 | 0.032 | 0.338 | **0.012** | **0.004** | 0.335 |
| | *DynamicGen*(Ours) | **0.108** | **0.029** | **0.282** | **0.012** | **0.004** | **0.331** |

## 4.3 Camera Intrinsics Estimation

Camera intrinsics are typically unavailable for most casual videos, especially those sourced from the Internet. However, accurate intrinsics are critical for reliable pose estimation and 3D reconstruction. To assess this, we evaluate focal length estimation accuracy on the Sintel dataset, with results summarized in Tab. 4. UniDepth predicts depth and focal length from a single image, while Dust3r processes sequential frames but is trained under classical multi-view settings and fails to generalize well to dynamic scenes. In contrast, *DynamicGen* demonstrates strong generalization to dynamic content and achieves the best performance in both Absolute Focal Error (AFE) and Relative Focal Error (RFE), setting a new state-of-the-art for focal length estimation in unconstrained video scenarios.

Table 4: **Camera intrinsics estimation.**

| Method | $AFE_{(px)}$↓ | $RFE_{(\%)}$↓ |
|---|---|---|
| UniDepth [39] | 447.4 | 0.357 |
| Dust3r [41] | 434.0 | 0.364 |
| *DynamicGen*(Ours) | **413.1** | **0.241** |

Table 5: **Dynamic Scene Caption evaluation.**

| Method | Acc.↑ | Com.↑ | Con.↑ | Rel.↑ | Avg.↑ |
|---|---|---|---|---|---|
| DO | 79.28 | 76.65 | 73.23 | 80.33 | 77.37 |
| + SAKFE | 80.23 | 77.46 | 74.01 | 81.45 | 78.29 |
| + HP | 82.57 | 81.42 | 71.17 | 82.56 | 79.43 |
| + Rephrasing | 82.48 | 80.50 | 71.86 | 83.27 | 79.53 |
| + COT | **84.38** | **82.09** | **75.87** | **85.56** | **81.97** |

## 4.4 Caption Quality Evaluation

To assess caption quality, we sampled 100 videos from the SA-V dataset [30]. As presented in Table 5, our experimental results indicate that integrating semantic-aware key frame extraction (SAKFE), hierarchical prompting (HP), caption rephrasing, and Chain-of-Thought (CoT) prompting [80] significantly enhances the quality of dynamic scene captions generated by Vision-Language Models (VLMs). We evaluated caption quality using the LLM-as-Judge metric G-VEval [81], conducting ten independent evaluations to ensure robust average results. The resulting captions were demonstrably more accurate, complete, concise, and relevant than those produced by direct output (DO), confirming the effectiveness of these strategies for improving caption quality in this task.

## 5 Conclusion

In this work, we address key limitations in traditional 4D data curation regarding scalability, physical realism, and modality diversity. We introduce DynamicGen, an automated pipeline leveraging foundation models for video filtering, metric-scale geometry and motion recovery, and hierarchical semantic captioning from raw videos. DynamicGen's capabilities are validated through standard benchmarks on video depth and camera pose/intrinsics estimation, qualitative analyses on diverse web videos, and human/LLM-based evaluations confirming caption quality. Utilizing DynamicGen, we construct DynamicVerse, a large-scale 4D dataset with over 100K dynamic scenes and rich physically grounded multimodal annotations. Together, this work offers a scalable 4D data generation methodology and a comprehensive new resource to advance 4D scene understanding.

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

Figure 6: DynamicVerse dataset.

# A Appendix

In the appendix, we provide more results and analysis and summarize them as follows:

- In Section A.1, we introduce the broader impact of our DynamicVerse framework.
- In Section A.2, we supplement details of dynamic bundle adjustment.
- In Section A.3, we ablate the different components for dynamic bundle adjustment.
- In Section A.4, we provide additional experiments on generated hierarchical captions.
- In Section A.5, we provide more qualitative results of dynamic bundle adjustment.
- In Section A.6, we provide inference speed and computational cost for DynamicGen.
- In Section A.7, we provide the limitation.

## A.1 Broader Impact

The introduction of DynamicVerse, with its large-scale, physically-aware, and multimodally annotated 4D dataset derived from real-world videos, is set to significantly influence several advanced research areas. Our framework's unique ability to capture metric-scale geometry, real-world motion, instance-level semantics, and descriptive captions offers an unparalleled resource that can catalyze progress in the following domains:

- **Dynamic 4D Scene Generation**: DynamicVerse offers a paradigm shift for Dynamic 4D Scene Generation. Current methods often rely on limited simulators or struggle to realistically portray complex real-world physics and motion from internet-sourced content. By accurately interpreting real-world dynamics from monocular videos and integrating window-based Bundle Adjustment with global optimization, DynamicVerse converts long video sequences into a comprehensive 4D multimodal format, capturing fine-grained dynamic information. This rich, real-world data provides an unparalleled training ground for generative models, leading to the creation of highly

realistic, physically plausible, and semantically coherent dynamic 4D scenes. This has profound implications for high-fidelity content creation in entertainment (e.g., movies, games), realistic virtual environments for training and simulation (e.g., disaster response, architectural visualization), and the synthetic generation of diverse data for further AI research, helping to overcome privacy and data collection limitations.

- **4D Vision-Language Models (4D-VLM)**: DynamicVerse will greatly accelerate the development of sophisticated 4D Vision-Language Models that can reason about space, time, and semantics concurrently. Existing VLMs often operate on 2D images or short video clips with limited 3D awareness. Our framework provides a unique combination of metric-scale 4D geometry, real-world dynamic motion, and comprehensive textual descriptions for long video sequences, allowing 4D-VLMs to learn intricate relationships between evolving 3D scenes and natural language narratives. Such models could enable more advanced human-agent interaction, where agents can provide detailed textual explanations of complex dynamic events they perceive in 4D, or understand nuanced, temporally extended instructions involving interactions within a 3D space. This could revolutionize areas like AI-powered video captioning, temporal question answering in 3D, and the development of embodied AI agents that communicate their understanding of the dynamic world with human-like richness.

- **4D Language-Grounded Gaussian Splatting (4D-LangSplat)**: DynamicVerse offers a foundational dataset for advancing 4D-LangSplat methodologies. While current 4D Gaussian Splatting techniques excel at novel view synthesis of dynamic scenes, their integration with language for semantic understanding and manipulation is still nascent. Our dataset, rich with 800K+ instance masks and holistic descriptive captions directly linked to evolving 3D structures and motions at a physical scale, empowers 4D-LangSplat models. This will enable the development of systems that can not only reconstruct dynamic scenes with high fidelity but also allow users to query, edit, and interact with these 4D representations using natural language. For instance, users could ask an agent to "track the red car that just turned left" or "remove the person walking in front of the fountain," with the model understanding both the spatial dynamics and the semantic context. This can significantly enhance applications in robotics, augmented reality, and interactive content creation by bridging the gap between visual perception and linguistic instruction in dynamic 3D environments.

In summary, DynamicVerse is poised to serve as a crucial catalyst, providing the data and framework necessary to bridge the gap between 2D understanding and true 4D world modeling, thereby fostering advancements in semantic scene understanding, dynamic object interaction, multimodal reasoning, and realistic content generation.

### A.2 Details of Dynamic Bundle Adjustment

**Camera parameterization**    In Eq. (1), $\boldsymbol{\xi} \in \mathbb{SE}(3)$ represents the camera poses as rigid transformations. Rotations are parameterized using $so(3)$ rotation vectors, which offer a minimal representation facilitating direct optimization.

**Static Area Bundle Adjustment Term**    In Eq. (2), the bundle adjustment energy $C_{\text{BA}}(\mathbf{P}, \mathbf{X}_{\text{static}})$ measures the consistency between the pixel-level correspondences and the 3D structure of static scene elements. Given the input pixel tracks $\mathcal{Z} = \{\mathbf{Z}_k\}_{k=0}^{K}$ and video segmentation $\mathcal{M} = \{\mathbf{M}^t\}_{t=0}^{T}$, we filter all tracks corresponding to static areas and minimize the distance between the projected pixel location and the observed pixel location:

$$C_{\text{BA}}(\mathbf{P}, \mathbf{X}_{\text{static}}; \mathcal{Z}, \mathcal{M}) = \sum_{\mathbf{Z}_k \in \mathcal{M}} \sum_{t} w_{k,t} \|\mathbf{Z}_{k,t} - \pi_K(\mathbf{X}_k, \boldsymbol{\xi}_t)\|_2 \tag{5}$$

where $\mathbf{X}_k$ is the $k$-th 3D point, $\mathbf{Z}_{k,t}$ is the $k$-th 3D point's corresponding pixel track's 2D coordinates at time $t$, $w_{k,t} \in \{0, 1\}$ is a visibility indicator and $\pi_K$ is the perspective projection function.

**Camera Smoothness Prior**    In Eq. (2), given the video input, a temporal smoothness prior is imposed on camera poses. This prior penalizes abrupt changes in relative pose, defined as $\xi_{t \to t+1} = \xi_{t+1}^{-1} \cdot \xi_t$. We adaptively reweight this term based on the magnitude of the relative motion. Specifically, a larger relative motion results in a reduced penalty on its change rate, while a smaller relative motion

incurs a higher penalty. Formally, this is expressed as:

$$C_{\text{cam}}(\mathbf{P}) = \sum_t C_{\text{rot}}(\mathbf{R}_{t-1,t,t+1}) + \sum_t C_{\text{trans}}(\mathbf{T}_{t-1,t,t+1})$$

where $C_{\text{rot}}(\mathbf{R}_{t-1,t,t+1}) = \frac{2||\text{rad}(\mathbf{R}_{t \to t+1}) - \text{rad}(\mathbf{R}_{t-1 \to t})||}{||\text{rad}(\mathbf{R}_{t-1 \to t})|| + ||\text{rad}(\mathbf{R}_{t \to t+1})||}$ and $C_{\text{trans}}(\mathbf{t}_{t-1,t,t+1}) = \frac{2||\mathbf{t}_{t \to t+1} - \mathbf{t}_{t-1 \to t}||}{||\mathbf{t}_{t-1 \to t}|| + ||\mathbf{t}_{t \to t+1}||}$; rad converts the rotation matrix into absolute radians.

**Non-Rigid Bundle Adjustment Term** In Eq. (3), for dynamic objects, we impose a nonrigid bundle adjustment term, $E_{\text{NR}}(\mathbf{X}_{\text{dyn}})$, which measures the discrepancy between the dynamic point cloud and pixel tracklets. Here, each pixel tracklet corresponds to a dynamic 3D point sequence, $\{\mathbf{X}_{k,t}\}$, optimized for each observed tracklet:

$$C_{\text{NR}}(\mathbf{X}_{\text{dyn}}, \mathbf{P}, \mathcal{Z}, \mathcal{M}) = \sum_{\mathbf{z}_k \in \mathcal{M}} \sum_t w_{k,t} \|\mathbf{Z}_{k,t} - \pi_K(\mathbf{X}_{k,t}, \boldsymbol{\xi}_t)\|_2 \tag{6}$$

where $\mathbf{X}_{k,t} \in \mathbb{R}^3$ is the $k$-th dynamic point's location at $t$.

**Dynamic Motion Prior** In Eq. (3), $C_{\text{motion}}(\mathbf{X}_{\text{dyn}})$ is a regularization term that encodes the characteristics of the dynamic structure. It contains two prior terms that are used to regularize the dynamic structure, both of which have demonstrated effectiveness in previous work.

$$C_{\text{motion}}(\mathbf{X}_{\text{dyn}}) = C_{\text{arap}}(\mathbf{X}_{\text{dyn}}) + C_{\text{smooth}}(\mathbf{X}_{\text{dyn}}). \tag{7}$$

$C_{\text{arap}}$ represents an as-rigid-as-possible (ARAP) prior [82] designed to penalize extreme deformations that compromise local rigidity. Specifically, for each dynamic control point $k$, its nearest neighbors are identified using k-Nearest Neighbors (KNN) on the remaining tracks. We then enforce that the relative distances among these neighboring pairs remain consistent, preventing sudden changes

$$C_{\text{arap}} = \sum_t \sum_{(k,m)} w_{km} \|d(\mathbf{X}_{k,t}, \mathbf{X}_{m,t}) - d(\mathbf{X}_{k,t+1}, \mathbf{X}_{m,t+1})\|^2 \tag{8}$$

where $d(,)$ is the L2 distance and $w_{km,t} = 1$ if all relevant points are visible.

$C_{\text{smooth}}$ is a simple smoothness term that promotes temporal smoothness for the dynamic point cloud:

$$C_{\text{smooth}} = \sum_{\mathbf{t}} \sum_{\mathbf{X}_k \in \mathbf{X}_{\text{dyn}}} w_{k,t} \|\mathbf{X}_{k,t} - \mathbf{X}_{k,t+1}\|_2. \tag{9}$$

Despite simplicity, both motion terms are crucial in our formulation, as they significantly reduce ambiguities in 4D dynamic structure estimation, which is highly ill-posed. Unlike other methods, we do not assume strong modelbased motion priors, such as rigid motion [83], articulated motion [84], or a linear motion basis [85].

**Optical Flow Prior** In Eq. (4), we also use a flow projection loss to encourage the global pointmaps to be consistent with the estimated flow for the confident, static regions of the actual frames. More precisely, given two frames $t, t'$, using their global pointmaps, camera extrinsics and intrinsics, we compute the flow fields from taking the global pointmap $\mathbf{X}_t$, assuming the scene is static, and then moving the camera from $t$ to $t'$. We denote this value $\mathbf{F}_{\text{cam}}^{\text{global: } t \to t'}$, similar to the term defined in the confident static region computation above. Then we can encourage this to be close to the estimated flow, $\mathbf{F}_{\text{est}}^{t \to t'}$, in the regions which are confidently static $\mathbf{X}_{\text{staic}}^{\text{global: } t \to t'}$ according to the global parameters:

$$C_{\text{flow}}(\mathbf{X}_{\text{static}}) = \sum_{W^i \in W} \sum_{t' \in W^i} \|\mathbf{X}^{\text{global: } t \to t'} \cdot (\mathbf{F}_{\text{cam}}^{\text{global: } t \to t'} - \mathbf{F}_{\text{est}}^{t \to t'})\|_1, \tag{10}$$

where $\cdot$ indicates element-wise multiplication. Note that the confident dynamic mask is initialized using the foundation models as described in Sec. 3.3. During the optimization, we use the global static pointmaps and camera parameters to compute $\mathbf{F}_{\text{cam}}^{\text{global}}$ and update the confident dynamic mask.

## A.3 Ablation Study on Different Components for Dynamic Bundle Adjustment

Our dynamic BA pipeline introduces three key components absent in prior work like Uni4D [55], which systematically improve the decomposition of static/dynamic elements and global consistency:

- **(a) Epi-Mask-Based Dynamics Filtering:** We introduce a geometric filtering step using an epipolar-based mask ("Epi-mask") to achieve a cleaner separation between static background and dynamic foreground pixels before bundle adjustment. This leads to more stable camera pose estimation and background reconstruction.

- **(b) VLM-Based Semantic Dynamics Analysis:** We leverage a Vision-Language Model (VLM) for a high-level, semantic understanding of motion. This enables intelligent, motion-aware keyframe extraction and provides robust masks for dynamic objects, a significant improvement over purely geometric or flow-based segmentation.

- **(c) Optical Flow-Based Sliding Window Global Refinement:** To address error accumulation and temporal drift common in long videos, we implement a global refinement strategy over a sliding window. This enforces long-range temporal consistency, correcting errors that a frame-by-frame or local BA approach would miss.

Table 6: Components Ablation on Sintel.

| Ablations | (a) | (b) | (c) | ATE$\downarrow$ | RPE$_{trans}\downarrow$ | RPE$_{rot}\downarrow$ | Abs$\downarrow$ | $\delta1.25\uparrow$ |
|---|---|---|---|---|---|---|---|---|
| Baseline | | | | 0.114694 | 0.032125 | 0.347920 | 0.216433 | 0.725167 |
| Ablation-1 | ✓ | | | 0.114065 | 0.032250 | 0.335198 | 0.215058 | 0.726943 |
| Ablation-2 | | ✓ | | 0.11053 | 0.033122 | 0.334005 | 0.210339 | 0.722999 |
| Ablation-3 | | | ✓ | 0.114694 | 0.032125 | 0.347920 | 0.214282 | 0.724084 |
| Ablation-4 | ✓ | ✓ | | 0.108459 | 0.028906 | 0.281979 | 0.205892 | 0.727616 |
| Ablation-5 | ✓ | | ✓ | 0.114065 | 0.032250 | 0.335198 | 0.214143 | 0.725534 |
| Ablation-6 | | ✓ | ✓ | 0.110530 | 0.033122 | 0.334005 | 0.207329 | 0.725784 |
| *DynamicGen* (Ours) | ✓ | ✓ | ✓ | **0.108459** | **0.028906** | **0.281979** | **0.204574** | **0.728961** |

## A.4 Additional experiments on generated hierarchical captions.

We performed three distinct experiments to validate the high quality of our hierarchical semantic annotations:

- **(a) Object-Level Semantics via 4D-LangSplat [67]:** To validate the annotations produced by our DynamicGen framework, we performed a time-sensitive querying experiment using a 4D-LangSplat model. For this evaluation, we trained the model on the "americano" scene from the HyperNeRF dataset and benchmarked it against a re-implemented 4D-LangSplat* baseline. The results, presented in Tab. 7, demonstrate that our approach yields substantial gains in Accuracy and volumetric Intersection over Union (vIoU). This superior performance confirms that our precise object masks and labels are highly effective for demanding multi-modal applications.

Table 7: Quantitative comparisons of time-sensitive querying on the HyperNeRF [86] dataset.

| Method | americano | |
|---|---|---|
| | Acc(%) | vIoU(%) |
| 4D-LangSplat* [67] | 53.84 | 27.55 |
| DynamicGen | **64.42** | **51.65** |

- **(b) Scene-Level Semantics via G-VEval [81]:** To rigorously assess our scene-level captions, we moved beyond single-score metrics and employed a more granular evaluation using the ACCR framework in G-VEval benchmark. This approach provides a comprehensive, multi-dimensional assessment of caption quality across four key axes: Accuracy, Completeness, Conciseness, and Relevance. On a random sample of 100 videos from SA-V data, our generated captions demonstrated high performance across all four criteria, as detailed in the Tab. 8. The strong

performance across these metrics confirms that our captions are not only factually accurate and relevant to the video content, but also complete in their coverage of events and efficiently concise. This robust, multi-faceted quality makes them highly suitable and reliable for demanding downstream applications.

Table 8: Evaluation of generated captions using the ACCR framework from G-VEval.

| Evaluation Criteria | Accuracy↑ | Completeness↑ | Conciseness↑ | Relevance↑ | Average↑ |
|---|---|---|---|---|---|
| Scene-Level Captions | 84.38 | 82.09 | 75.87 | 85.56 | 81.97 |

- **(c) Camera-Level Semantics via Human Study:** We conducted a formal human study to quantitatively analyze the quality of the final camera motion captions. Following prior work [71], we asked human evaluators to rate our captions on three criteria: (1) Clearness (clarity of information), (2) Conciseness (brevity without losing clarity), and (3) Grammar & Fluency. On a sub-sample of 88 videos from our dataset (i.e., filtered DAVIS), our captions performed excellently. The results, presented in Tab. 9 showed that over 60.22% of the captions were rated as both clear and fluent, while also receiving high scores for conciseness. This confirms the effectiveness of our generation and quality control process.

Table 9: Human evaluation results for the generated camera captions. Scores indicate the percentage of captions that met each quality criterion.

| Human Evaluation | Rated as Clear | Rated as Fluent | Rated as Concise |
|---|---|---|---|
| Camera Captions | 85.22% | 89.77% | 67.04% |

## A.5 More qualitative results of dynamic bundle adjustment

We present additional qualitative reconstruction results in Fig. 8, demonstrating the generalizability and performance of our pipeline on real-world data.

## A.6 Inference Speed and Computational Cost for DynamicGen

For a reproducible analysis of computational performance, we processed the entire Sintel training set (23 videos) on NVIDIA H20 GPUs. A detailed breakdown of the average processing time and peak VRAM consumption for each component of our pipeline is provided in Table 10.

Table 10: Computational Cost Analysis.

| Module | Hardware Used | Avg. Time / Sintel Video (mins) | Peak VRAM (GB) | Notes |
|---|---|---|---|---|
| 1. Motion-aware Keyframe Extraction | 1x H20 GPU | ∼0.1 | ∼10 | Selects representative frames |
| 2. VLM-Based Semantic Analysis (Qwen-VL) | 2x H20 GPU | ∼1.6 | ∼60 | Identifies dynamic elements |
| 3. Moving Object Segmentation (SA2VA) | 1x H20 GPU | ∼0.8 | ∼30 | Per-object video segmentation |
| 4. Dynamic Bundle Adjustment | 1x CPU Core + 1x H20 GPU | ∼12.2 | ∼30 | Main time bottleneck |
| 5. Moving Object Captioning | 2x H20 GPU | ∼2.0 | ∼24 | Object-level descriptions |
| 6. Dynamic Scene Captioning | 2x H20 GPU | ∼3.0 | ∼40 | Scene-level descriptions |
| 7. Camera Motion Captioning | 2x H20 GPU | ∼2.0 | ∼40 | Camera-level descriptions |
| 8. Caption Rephrasing | 1x H20 GPU | ∼2.0 | ∼24 | LLM-based refinement for consistency and conciseness |
| **Total (per video)** | **H20 GPU** | **∼23.7** | **∼60** | **Peak VRAM, not sum** |

## A.7 Limitations

Despite its considerable capabilities, DynamicVerse exhibits several inherent limitations. First, its reliance on in-the-wild internet videos introduces significant noise and quality variance. This can compromise the fidelity of metric-scale geometry and motion recovery, particularly in complex, cluttered, or occluded scenes that fall outside the typical distribution of the foundation models' training

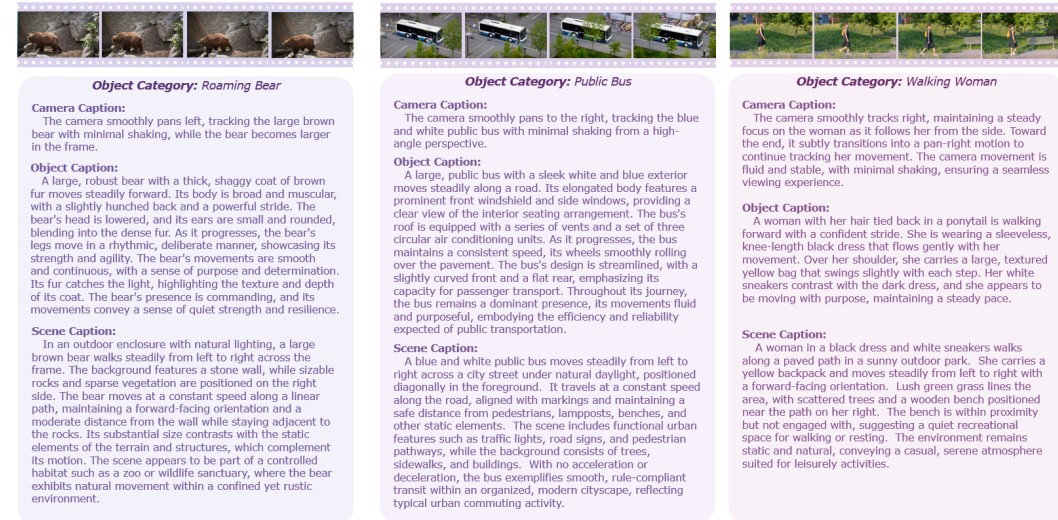

**Object Category:** *Roaming Bear*

**Camera Caption:**
The camera smoothly pans left, tracking the large brown bear with minimal shaking, while the bear becomes larger in the frame.

**Object Caption:**
A large, robust bear with a thick, shaggy coat of brown fur moves steadily forward. Its body is broad and muscular, with a slightly hunched back and a powerful stride. The bear's head is lowered, and its ears are small and rounded, blending into the dense fur. As it progresses, the bear's legs move in a rhythmic, deliberate manner, showcasing its strength and agility. The bear's movements are smooth and continuous, with a sense of purpose and determination. Its fur catches the light, highlighting the texture and depth of its coat. The bear's presence is commanding, and its movements convey a sense of quiet strength and resilience.

**Scene Caption:**
In an outdoor enclosure with natural lighting, a large brown bear walks steadily from left to right across the frame. The background features a stone wall, while sizable rocks and sparse vegetation are positioned on the right side. The bear moves at a constant speed along a linear path, maintaining a forward-facing orientation and a moderate distance from the wall while staying adjacent to the rocks. Its substantial size contrasts with the static elements of the terrain and structures, which complement its motion. The scene appears to be part of a controlled habitat such as a zoo or wildlife sanctuary, where the bear exhibits natural movement within a confined yet rustic environment.

**Object Category:** *Public Bus*

**Camera Caption:**
The camera smoothly pans to the right, tracking the blue and white public bus with minimal shaking from a high-angle perspective.

**Object Caption:**
A large, public bus with a sleek white and blue exterior moves steadily along a road. Its elongated body features a prominent front windshield and side windows, providing a clear view of the interior seating arrangement. The bus's roof is equipped with a series of vents and a set of three circular air conditioning units. As it progresses, the bus maintains a consistent speed, its wheels smoothly rolling over the pavement. The bus's design is streamlined, with a slightly curved front and a flat rear, emphasizing its capacity for passenger transport. Throughout its journey, the bus remains a dominant presence, its movements fluid and purposeful, embodying the efficiency and reliability expected of public transportation.

**Scene Caption:**
A blue and white public bus moves steadily from left to right across a city street under natural daylight, positioned diagonally in the foreground. It travels at a constant speed along the road, aligned with markings and maintaining a safe distance from pedestrians, lampposts, benches, and other static elements. The scene includes functional urban features such as traffic lights, road signs, and pedestrian pathways, while the background consists of trees, sidewalks, and buildings. With no acceleration or deceleration, the bus exemplifies smooth, rule-compliant transit within an organized, modern cityscape, reflecting typical urban commuting activity.

**Object Category:** *Walking Woman*

**Camera Caption:**
The camera smoothly tracks right, maintaining a steady focus on the woman as it follows her from the side. Toward the end, it subtly transitions into a pan-right motion to continue tracking her movement. The camera movement is fluid and stable, with minimal shaking, ensuring a seamless viewing experience.

**Object Caption:**
A woman with her hair tied back in a ponytail is walking forward with a confident stride. She is wearing a sleeveless, knee-length black dress that flows gently with her movement. Over her shoulder, she carries a large, textured yellow bag that swings slightly with each step. Her white sneakers contrast with the dark dress, and she appears to be moving with purpose, maintaining a steady pace.

**Scene Caption:**
A woman in a black dress and white sneakers walks along a paved path in a sunny outdoor park. She carries a yellow backpack and moves steadily from left to right with a forward-facing orientation. Lush green grass lines the area, with scattered trees and a wooden bench positioned near the path on her right. The bench is within proximity but not engaged with, suggesting a quiet recreational space for walking or resting. The environment remains static and natural, conveying a casual, serene atmosphere suited for leisurely activities.

Figure 7: Examples captions on DAVIS dataset.

data. Second, the substantial computational overhead required to process long video sequences with large-scale models presents a practical barrier to real-time performance and scalable deployment. Finally, while extensive, the dataset cannot exhaustively capture the long tail of real-world phenomena. Consequently, the model's generalization to truly novel environments is fundamentally tethered to the intrinsic biases and capabilities of its underlying foundation models.

These limitations raise AI-safety concerns: (i) privacy and security risks, since metric-scale reconstructions from web videos can expose sensitive interiors or critical infrastructure and facilitate covert mapping or surveillance; and (ii) miscalibrated confidence under distribution shift, producing plausible but erroneous geometry and dynamics that misguide downstream robotic or AR planners. Biases and licensing gaps in foundation models and web data may further perpetuate representational harms and legal or IP issues. A practical mitigation is to prefilter ineligible videos using policy rules and automated detectors (e.g., content with PII, sensitive interiors or infrastructure, minors, or restricted licenses).

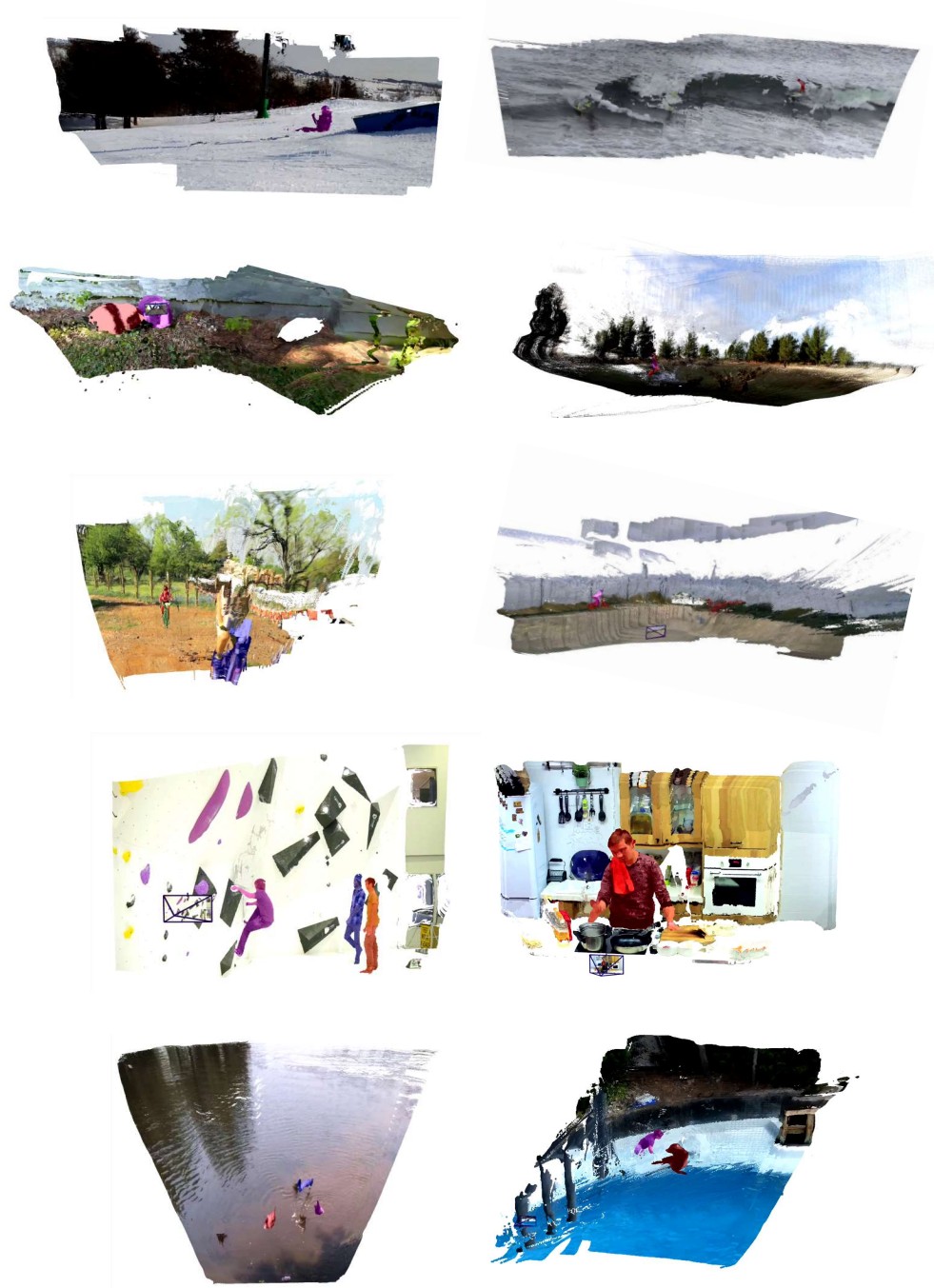

Figure 8: **Qualitative Results on in-the-wild data**. We show qualitatively some of our reconstruction results on in-the-wild data. For full reconstruction, please refer to our attached supplementary webpage.

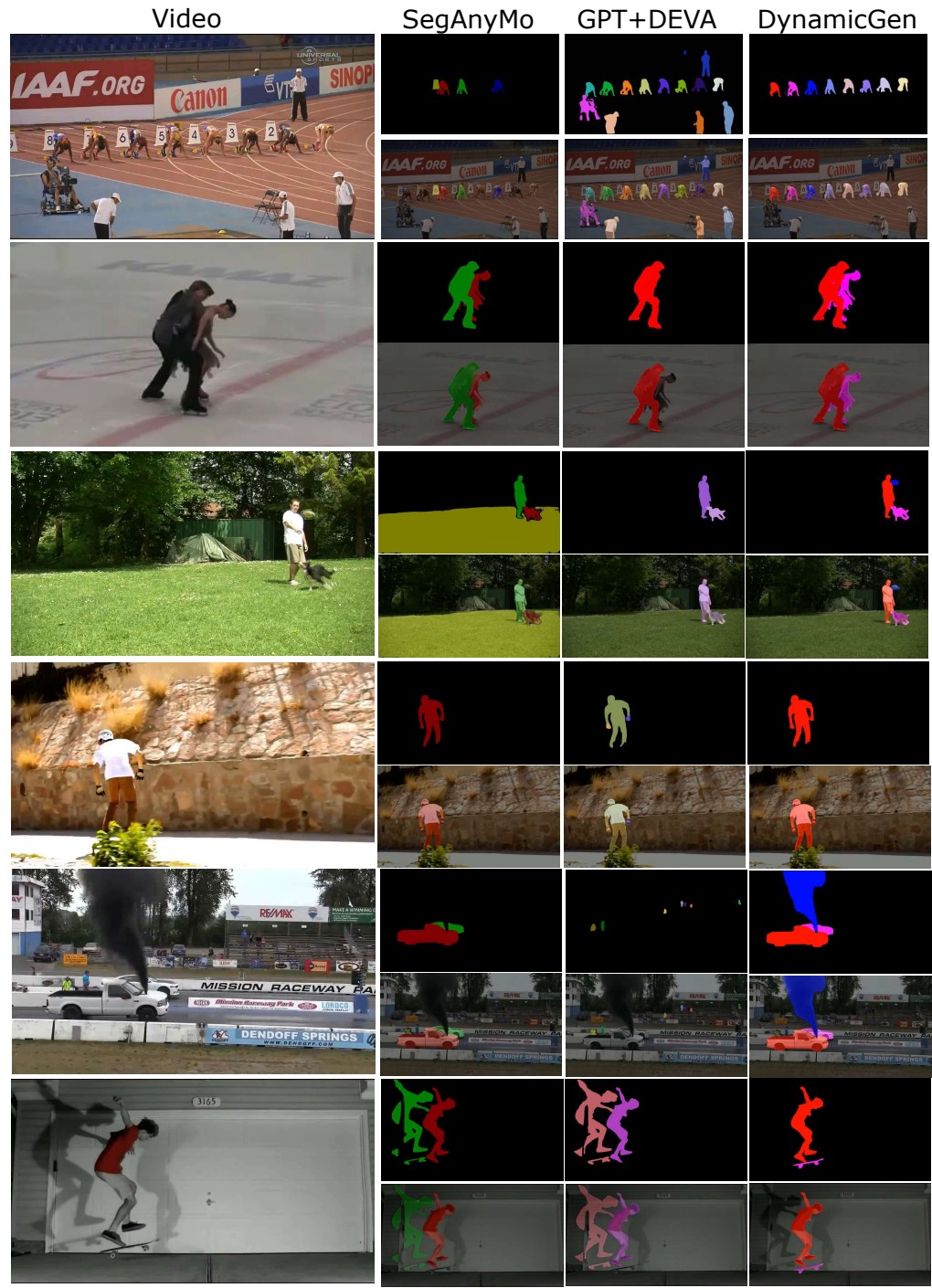

Figure 9: **Qualitative Results of moving object Segmentation**. We show qualitatively some of our segmentation results on the Youtube-VIS dataset compared with other baselines.

