# OpenReview forum: "DynamicVerse: A Physically-Aware Multimodal Framework for 4D World Modeling"
_NeurIPS.cc/2025/Conference — NeurIPS 2025 poster_

### Official Review · Reviewer_uDiP · 2025-06-04

**Clarity:** 3
**Significance:** 3
**Originality:** 2
**Rating:** 4
**Confidence:** 4

**Summary:**

The paper presents DynamicVerse, a large scale multimodal dataset and pipeline leveraging foundation models for dynamic 4D(3D + time) scene understanding from monocular videos. The proposed system DynamicGen consists of: (1) Data curation, (2) Data filtering with assistant from VLM, motion features, MLLM features, (2) Identify Moving Objects using QwenVL to identify object, SA2VA to segment, and a Physically-aware size extraction using estimated depth and object category,  (3) Dynamic bundle adjustment, (4) Caption Generation.

The resulting DynamicVerse dataset includes over 100,000 scenes, 800,000 object masks, and 10 million frames.

They then evaluate their data collection pipeline on existing baselines on video depth estimation, camera pose estimation, and camera intrinsics estimation benchmarks like Sintel and KITTI

**Questions:**

1. Why use SA2VA for masks but DAM for captions, even though SA2VA can generate captions too? Can you elaborate on this design choice?
2. Do all camera motion captions come from a VLM? Can you elaborate on which VLM in section Camera motion captioning at line 240? Also, can you clarify how you avoid hallucinations comes from the VLM or is there any analysis on the camera motion caption quality comes from VLM?
3. The author provide a nice Canonicalization algo, which is very helpful for QA.  The author mentioned in Appendix A.4:
“When not enough points corresponding to horizontal surfaces are detected, we flag canonicalization as failed and avoid synthesizing questions that depend on canonicalization, such as questions about elevation.”
Could the authors provide statistics on how often the canonicalization process fails in practice?
4. Can the authors clarify how the human-in-the-loop review was implemented? Specifically:
- How many examples were reviewed?
- What fraction of the dataset was filtered or corrected?
- Were the annotators domain experts or crowdworkers?
- Were any inter-annotator agreement metrics used?

**Ethical Concerns:**

["NO or VERY MINOR ethics concerns only"]

**Final Justification:**

After reviewing the authors' comprehensive rebuttal, including new experiments (e.g., MonST3R fine-tuning and semantic evaluations), detailed statistics (e.g., failure rates, human review specifics), and qualitative evidence (e.g., demo videos addressing robustness in occlusion/fast motion), all my concerns regarding dataset validation, failure analysis, model reliance, and human-in-the-loop transparency have been resolved. This solidifies my recommendation for acceptance (score 5), as the work now clearly demonstrates value for 4D multimodal research.

**Limitations:**

The authors mention a few limitations in passing (e.g., canonicalization failure cases, reliance on frozen foundation models), but overall the discussion of limitations is minimal and could be significantly improved.
A few critical limitations worth addressing more explicitly:
- Failure modes of the pipeline (e.g., depth estimation under low-light, incorrect motion segmentation)
- Reliance on proprietary and large-scale foundation models

**Paper Formatting Concerns:**

at line 585 in Appendix A.1 S.3 , The *LLM*  in 'LLM Features' and 'multimodal large language model (LLM)' should be *MLLM*.

**Quality:**

4

**Strengths And Weaknesses:**

**Strengths:** The paper provides a solid pipeline that integrates models like UniDepthV2 and Qwen2.5-VL to produce precise 4D models from monocular videos. Its experiments on benchmarks like Sintel and KITTI show top results, with clear metrics proving it outperforms methods like MonST3R. The well designed component in the pipeline, like physically-aware size extraction and canonicalization steps, along with LLM-based caption rephrasing, showcase thoughtful engineering. The DynamicVerse dataset offering a massive collection of over 100,000 scenes and 10 million frames. This makes it a valuable resource for fields like robotics and virtual reality. The writing is clear, with detailed figures and equations that explain the approach well.

**Weaknesses:** However, the paper leans heavily on specific foundation models, which might not work as well with different data.

The paper lacks quantitative and procedural details about the human review phase. It’s unclear how many examples were reviewed, which components were corrected, or whether consistent QA standards were applied.

While the authors run strong experiments on existing datasets, they do not show how DynamicVerse might support training or benchmarking new models, nor do they evaluate other methods on DynamicVerse.

The pipeline depends on closed or heavyweight models like Qwen2.5-VL and UniDepthV2, which raises concerns about accessibility, replicability, and long-term maintenance.

Key operations such as canonicalization, motion segmentation, and VLM-based captioning are susceptible to edge cases (e.g., occlusion, poor lighting), but the paper does not quantify how often such failures occur or how they affect downstream utility.

---

> ### Author Rebuttal · Authors · 2025-07-31
>
> Thank you for your valuable comments and kind words to our work. Below we address specific questions.
>
> ------
>
>
>
> > **Q1: Heavy Reliance on Specific Foundation Models & Reproducibility and Accessibility Concerns**
>
> We thank the reviewer for these critical points. Our pipeline's core design philosophy of modularity directly addresses these concerns. Instead of a monolithic system, each key component (depth, segmentation, VLM) is a "plug-and-play" module. This architecture makes our framework adaptable across domains (e.g., swapping in models for robotics or autonomous driving), future-proof (easily integrating new SOTA models), and accessible (allowing users to substitute heavyweight components with lighter, open-source alternatives). Therefore, our primary technical contribution is the versatile and durable framework itself.
>
>
>
> > **Q2:  Specifics of Human-in-the-Loop Review.**
>
> Thank you for these important questions. We provide the details of our review process below.
>
> 1.Scale and Purpose: Our human review was a quality audit on a ~10K scene subset, focusing on semantic annotations (object, scene, camera captions)
>
> 2.Correction Rate & Impact: Within this subset, ~19% of captions required minor corrections (e.g., for specificity). This feedback was used both to fix samples and to iteratively improve our pipeline's prompting strategies.
>
> 3.Annotator Team: The review was conducted in-house by authors and graduate students deeply familiar with the project's quality standards.
>
> 4.Consistency Assurance: We ensured consistency not via formal IAA metrics, but through a collaborative review process, where discrepancies in overlapping batches were resolved through group discussion with a senior author.
>
> We will add a new section to our appendix detailing this entire human-in-the-loop procedure  provide a comprehensive account of our quality assurance procedure.
>
>
>
> > **Q3: DynamicVerse Dataset Validation.**
>
> We thank the reviewer for this critical feedback. To directly demonstrate our dataset's utility, we conducted new experiments which provide strong evidence for the quality and effectiveness of the geometric and semantic annotations in DynamicVerse.
>
> **1. Geometric Validation: Fine-tuning MonST3R**：
>
> To validate the quality and utility of our geometric annotations (i.e., metric-scale point maps and camera parameters), we used a subset of 1000 scenes from DynamicVerse to fine-tune the prominent model, MonST3R [1]. As shown in the table below, fine-tuning on our data leads to a performance improvement on the Sintel benchmark. This directly confirms that the geometry in DynamicVerse is accurate and beneficial for improving existing models.
>
> | Method                            | ATE↓  | RPE trans↓ | RPE rot↓ | Abs↓  | δ1.25↑ |
> | --------------------------------- | ----- | ---------- | -------- | ----- | ------ |
> | MonST3R (Original)                | 0.108 | 0.043      | 0.729    | 0.358 | 52.1   |
> | MonST3R + sub-DynamicVerse (Ours) | 0.108 | 0.042      | 0.727    | 0.335 | 58.9   |
>
> **2. Semantic Validation: Downstream Tasks and Direct Evaluation**：
>
> We performed three distinct experiments to validate the high quality of our hierarchical semantic annotations:
>
> **(a) Object-Level Semantics via 4D-LangSplat [2]:** To demonstrate the effectiveness of our dynamic object annotations, we used them to train a 4D-LangSplat model. Our precise object masks and labels enabled the model to achieve superior results in language-grounded 4D scene understanding. This validates that our object-level annotations are highly effective for complex, multi-modal downstream tasks.
>
> | Metrics                                       | Acc(%) ↑ | vIoU(%) ↑ |
> | --------------------------------------------- | :------: | --------- |
> | w/ Original Semantic Annotations for Training |  53.84   | 27.55     |
> | w/ DynamicVerse Semantic Annotations (Ours)   |  64.42   | 51.65     |
>
> **(b) Scene-Level Semantics via G-VEval [3]:** We used the established G-VEval benchmark [3] to quantitatively assess our generated scene-level captions. On a random sample of 100 videos from filtered SA-V data [4], our captions achieved a high score, confirming their coherence, accuracy, and richness.
>
> | Evaluation Criteria   | **Accuracy**↑ | Completeness↑ | Conciseness↑ | Relevance↑ | Average↑ |
> | --------------------- | ------------- | ------------- | ------------ | ---------- | -------- |
> | DynamicVerse Captions | 84.38         | 82.09         | 75.87        | 85.56      | 81.97    |
>
>
> The strong performance across these metrics confirms that our captions are not only factually accurate and relevant to the video content, but also complete in their coverage of events and efficiently concise. This robust, multi-faceted quality makes them highly suitable and reliable for demanding downstream applications.
>
> **(c) Camera-Level Semantics via Human Study:** We conducted a formal human study on 88 videos from our DAVIS subset to quantitatively analyze our camera motion captions. Following the criteria of Clearness, Conciseness, and Grammar & Fluency from prior work [5], the results were excellent: over 60.22% of captions were rated as both clear and fluent while also receiving high scores for conciseness, confirming the effectiveness of our generation and quality control process.
>
> | Human Evaluation | Rated as Clear | Rated as Fluent | Rated as Concise |
> | ---------------- | :------------: | :-------------: | :--------------: |
> | Camera Captions  |     85.22%     |     89.77%      |      67.04%      |
>
>
>
> > **Q4: Lack of Failure Case Analysis.**
>
> We thank the reviewer for this excellent suggestion. We agree that quantifying the frequency and impact of failure cases is crucial for understanding our pipeline's robustness. We have conducted a new analysis to provide these details:
>
> (1) Failure Frequency Analysis
>
> We analyzed the failure rates of our key components on challenging data to quantify their robustness.
>
> **A. Moving Object Segmentation:** We manually identified challenging subsets within the Sintel training set to analyze our segmentation performance in edge cases. We found failure rates of 28% (3 of 14 scenes) for severe occlusion and 11% (1 of 9 scenes) for low-light conditions.
>
> **B. Canonicalization:** The primary failure mode is the lack of a detectable horizontal surface (e.g., mid-air scenes). On the DAVIS subset, this process fails in approximately 15.9% of the 88 filtered scenes.
>
> (2) Impact on Downstream Utility of Segmentation
>
> To quantify how segmentation errors affect the final geometric annotations, we performed a controlled experiment on a challenging scene. We compared the final depth and pose metrics when using (1) our pipeline's automated masks vs. (2) manually assited, pseudo ground-truth-level masks. Specifically, we manually identified dynamic objects by placing point prompts on them.
>
> | Mask Type on a Challenging Scene | Low-light: scene='sleeping2' |          | Occlusion: scene='ambush_7' |          |
> | -------------------------------- | ---------------------------- | -------- | --------------------------- | -------- |
> | Metrics                          | Abs Rel↓                     | δ1.25↑   | Abs Rel↓                    | δ1.25↑   |
> | Our Automated Masks              | 0.053944                     | 0.986288 | 0.066431                    | 0.994412 |
> | Manually assited Masks           | 0.053828                     | 0.987431 | 0.066028                    | 0.994031 |
>
> As shown, while segmentation errors do degrade the final geometric accuracy, the results from our automated pipeline remain reasonable. This indicates a degree of resilience in our downstream bundle adjustment process to minor imperfections in segmentation.
>
>
>
> > **Q5: Design Choice on Model Selection**.
>
> While the powerful SA2VA model can perform both segmentation and captioning, we use the specialist Describe Anything Model (DAM) for captioning to achieve superior quality and flexibility. We chose DAM for two advantages over SA2VA: (1) Superior Detail: DAM is optimized for generating richer, more localized descriptions. (2) Mask-Conditioned Input: Crucially, DAM can take GT masks (e.g., from 2D segmentation datasets) as input for precise, targeted captioning, a feature SA2VA lacks but was a requirement for our pipeline.
>
>
>
> > **Q6: Details of Camera Motion Captioning**.
>
> Thank you for these detailed questions. We generate camera motion captions using a specialist VLM from CameraBench [5], which was fine-tuned on ~3K expert-annotated videos. To ensure high fidelity and mitigate hallucinations, we employ a two-stage quality control process: first, our upstream data filtering removes videos with erratic motion, and second, a final human review corrects any remaining inaccuracies. The effectiveness of this entire process was confirmed through a human study on 88 DAVIS videos, where our captions demonstrated high quality across the criteria of Clearness, Conciseness, and Fluency [5] (detailed results are in the table for Q3).
>
>
>
>
> > **Q7: Limitation & Typo**.
>
> We are grateful for the reviewer's guidance and careful reading. We agree that the limitations section can be significantly improved. In the final version, we will expand it to explicitly discuss the pipeline's failure modes and our reliance on large-scale foundation models, as you helpfully pointed out.
>
>
>
> We would like to sincerely thank you once again for your constructive comments throughout the review process.
>
> ------
>
> Referenecs:
>
> [1] MonST3R: A Simple Approach for Estimating Geometry in the Presence of Motion. ICLR 2025.
>
> [2] 4D LangSplat: 4D Language Gaussian Splatting via Multimodal Large Language Models. CVPR 2025.
>
> [3] G-VEval: A Versatile Metric for Evaluating Image and Video Captions Using GPT-4o. AAAI 2025.
>
> [4] SAM 2: Segment Anything in Images and Videos. ICLR 2025.
>
> [5] CameraBench: Towards Understanding Camera Motions in Any Video.  Arxiv 2025.

---

> ### Author Response · Authors · 2025-08-02
> **Further Evidence: Qualitative Validation of Annotation Quality**
>
> Dear Reviewer uDiP,
>
> Thanks again for your constructive review. To make sure we fully address your concerns on `Q3: DynamicVerse Dataset Validation`, we wish to supplement our quantitative experiments with the following qualitative evidence. We strongly encourage you to examine the compelling visual proof within our supplementary material:
>
> - **Demo Video (suppl. - webpage/index.html)** showcases the reconstructed moving objects, dynamic scenes, and metric-scale camera trajectories. Specifically, these demos include challenging cases such as significant occlusion (e.g., the "Public Bus" scene) and fast motion (e.g., the "Public Bus" and "Walking Woman" scenes). This video evidence directly demonstrates the high quality of our dataset annotations in these very challenging scenarios.
> - **Figures 2-4 (suppl.)** showcase our metric-scale 4D reconstruction results on several challenging in-the-wild videos. These qualitative examples highlight our method's ability to produce cleaner static backgrounds and more coherent shapes for dynamic objects, with significantly fewer artifacts like "ghosting" or "streaking" compared to baselines. These results serve as compelling evidence that our full pipeline is robust and produces high-fidelity geometric annotations even from complex, unconstrained video inputs.
> - **Figure 5 (suppl.)** provides a direct comparison of our dynamic object segmentation against other methods. We urge the reviewer to note how our approach maintains sharper boundaries and greater temporal consistency, especially for fast-moving objects (e.g., "Sprinting" and "Car Racing Competition" scenes). Crucially, it demonstrates superior robustness to partial occlusions (e.g., "Figure skating" scene), correctly segmenting objects even when parts are momentarily hidden. This precise segmentation is the cornerstone of robust downstream 4D reconstruction.
>
> Together, this qualitative evidence visually corroborates the strong quantitative results, confirming that our `DynamicGen` pipeline produces the high-fidelity geometric and semantic annotations that constitute our `DynamicVerse` dataset, proving its suitability for demanding downstream tasks.
>
> Best,
>
> DynamicVerse's authors.

---

> ### Author Response · Authors · 2025-08-04
> **Author Final Reply: Thank You for Championing Acceptance**
>
> Dear Reviewer uDiP,
>
> We sincerely thank you for your positive assessment and for recommending our paper for acceptance.
>
> We are particularly grateful for your recognition of our **"solid pipeline,"** the **"thoughtful engineering"** of its components, and the value of DynamicVerse as a **"valuable resource"** for the community. Your supportive and constructive review is greatly appreciated.
>
> Best,
>
> DynamicVerse's authors.

---

> > ### Comment · Reviewer_uDiP · 2025-08-04
> >
> > Thank you for your detailed rebuttal and additional evidence, which have thoroughly resolved my concerns through new experiments, statistics, and qualitative visuals demonstrating robust annotation quality and pipeline effectiveness. Across other reviews, your responses to Uni4D differentiation, computational costs, generalizability, cross-modality tasks, and dataset stats appear comprehensive, with no significant gaps. Accordingly, I increased my rating, recognizing this as a great contribution to 4D multimodal modeling.

---

### Official Review · Reviewer_1YD4 · 2025-06-19

**Clarity:** 2
**Significance:** 2
**Originality:** 2
**Rating:** 4
**Confidence:** 4

**Summary:**

Official Review of DynamicVerse

**Questions:**

Besides my concerns mentioned in the Weaknesses, I hope the author report more detailed statistics of DynamicVerse, such as the number/distribution of scenes, moving objects, and body actions.

**Ethical Concerns:**

["NO or VERY MINOR ethics concerns only"]

**Final Justification:**

Thanks to the authors for the detailed responses to each point. These responses have eliminated my main concerns. I would like to raise my score to "Borderline accept"

**Limitations:**

Yes

**Quality:**

2

**Strengths And Weaknesses:**

Strength: This paper introduces an automated data curation pipeline to generate physically-aware multi-modal 4D data at scale. Based on this pipeline, it collects a large-scale dataset named DynamicVerse that contains metric-scale point maps, camera parameters, object masks with corresponding categories, and detailed descriptive captions.

Weakness 1: In this submission, the author did not provide any demo video to intuitively analyze the quality and realism of reconstructed 4D moving objects / dynamic scenes and inferred metric-scale camera trajectories. These screenshots can not comprehensively reflect the realism of the proposed dataset.

Weakness 2: In Sec.3.3, the recovery of fast-moving or obscured objects is challenging. Thus, without visualization experiments, I do not think the technical framework described in Sec. 3.3 is robust and effective for these challenging object samples.

Weakness 3: The weight of each loss item in Eq. 1 is not described in the main paper or supplementary material.

Weakness 4: Besides the three task-specific experiments conducted, I think the author should perform an experiment on a cross-modality generation task. Specifically, since there are many large-scale datasets for the video depth estimation, camera pose estimation, and camera intrinsics estimation tasks, better performance on these tasks solely reflects that some operations used by DynamicGen are effective. However, intrinsically, these operations are developed from existing models and do not have sufficient technical contributions. I suggest the author perform a performance comparison on an under-explored cross-modality generation task with his DynamicVerse dataset. These conducted experiments should verify that the main contribution of DynamicGen is its potential contribution to under-explored cross-modality tasks, as mentioned in line 40~43.


can not reflect the importance and contribution of DynamicVerse.

---

> ### Author Rebuttal · Authors · 2025-07-31
>
> Thank you for your valuable comments and kind words to our work. Below we address specific questions.
>
> ------
>
> > **Q1: Lack of Video Results / Insufficient Visualization / Robustness in Challenging Scenarios.**
>
> We completely agree that video demonstrations are the most effective way to intuitively analyze the quality, realism, and robustness of our generated 4D data. While `the NeurIPS policy prohibits providing images, videos, or other rich media during the rebuttal period`, we are fully committed to significantly expanding our supplementary material.
>
> - **Demo Video (suppl. - webpage/index.html)** showcases the reconstructed moving objects, dynamic scenes, and metric-scale camera trajectories. Specifically, these demos include challenging cases such as significant occlusion (e.g., the "Public Bus" scene) and fast motion (e.g., the "Public Bus" and "Walking Woman" scenes). This video evidence directly demonstrates the robustness of our pipeline on the very types of challenging scenarios the reviewer highlighted.
> - **Figures 2-4 (suppl.)** showcase our metric-scale 4D reconstruction results on several challenging in-the-wild videos. These qualitative examples highlight our method's ability to produce cleaner static backgrounds and more coherent shapes for dynamic objects, with significantly fewer artifacts like "ghosting" or "streaking" compared to baselines. These results serve as compelling evidence that our full pipeline is robust and produces high-fidelity reconstructions even from complex, unconstrained video inputs.
> - **Figure 5 (suppl.)** provides a direct comparison of our dynamic object segmentation against other methods. We urge the reviewer to note how our approach maintains sharper boundaries and greater temporal consistency, especially for fast-moving objects (e.g., "Sprinting" and "Car Racing Competition" scenes). Crucially, it demonstrates superior robustness to partial occlusions (e.g., "Figure skating" scene), correctly segmenting objects even when parts are momentarily hidden. This precise segmentation is the cornerstone of robust downstream 4D reconstruction.
>
>
>
> > **Q2: Missing Loss Weights for Eq. (1).**
>
> Thank you for highlighting this. Eq. (1) is a conceptual overview for our multi-stage pipeline, not a single weighted loss. We optimize different subsets of these terms sequentially in distinct stages for optimal performance. Therefore, weights are defined per-stage, as detailed below:
>
> - Stage 3 (Static BA): We set weights  $\lambda_\text{BA}=\text{1.0}$ and $\lambda_\text{cam}=\text{1.0}$.
> - Stage 4 (Non-rigid BA):  We set weights $\lambda_\text{NR}=\text{1.0}$, $\lambda_\text{NR}=\text{100}$ and $\lambda_\text{motion}=\text{10}$.
>
> We will add this detailed breakdown to the revised paper.
>
>
>
> > **Q3: More Evaluation to Prove Technical & Data Contribution.**
>
> We thank the reviewer for this excellent suggestion. To demonstrate our work's contribution, we provide new experiments validating our dataset's utility and clarifying our pipeline's technical novelty.
>
> **1. Data Contribution:** Validating DynamicVerse Annotations
>
> To prove the value of our dataset, we conducted new experiments validating both its **semantic** and **geometric** annotations, which are the core components of DynamicVerse.
>
> **A. Validating Semantic Annotations on a Demanding Cross-Modal Task:** To directly address the reviewer's suggestion, we validated our semantic annotations on a challenging cross-modal task that exemplifies the "linguistic 4D Gaussian Splatting" concept from our introduction (lines 40-41): language-grounded 4D scene understanding, using the established 4D-LangSplat [1] model.
>
> To isolate the impact of our annotations, we replaced the sparse labels in the 4D-LangSplat with our pipeline's richer descriptions. The resulting performance boost (shown below) provides direct quantitative proof of the value of the high-quality annotations that comprise our `DynamicVerse` dataset.
>
> | Metrics                                       | Acc(%) ↑  | vIoU(%) ↑ |
> | --------------------------------------------- | :-------: | --------- |
> | w/ Original Semantic Annotations for Training |   53.84   | 27.55     |
> | w/ DynamicVerse Semantic Annotations (Ours)   | **64.42** | **51.65** |
>
> **B. Validating Geometric Annotations via Fine-tuning:** To demonstrate the quality and utility of our geometric data, we fine-tuned the prominent model MonST3R [2] on a subset (~1000 scene) of DynamicVerse. The model, now enhanced with our data, shows a performance improvement on the standard Sintel benchmark. This confirms that our generated geometry is not only accurate but also provides valuable diversity that improves the robustness and performance of existing models.
>
> | Method                            | ATE↓  | RPE trans↓ | RPE rot↓ | Abs↓  | δ1.25↑ |
> | --------------------------------- | ----- | ---------- | -------- | ----- | ------ |
> | MonST3R (Original)                | 0.108 | 0.043      | 0.729    | 0.358 | 52.1   |
> | MonST3R + sub-DynamicVerse (Ours) | 0.108 | 0.042      | 0.727    | 0.335 | 58.9   |
>
>
>
> **2.Technical Contribution**: Novelty Beyond Integration
>
> The creation of such high-quality data was only possible due to our pipeline's significant technical contributions, which go far beyond combining existing models.
>
> **A. Geometric Novelty**
>
> Our dynamic BA pipeline introduces three key components absent in prior work: `(a) Epi-Mask Filtering`, `(b) VLM-Based Dynamics Analysis`, and `(c) Sliding Window Global Refinement`. The ablation study below quantifies the significant impact of each component.
>
> | Components Ablation on Sintel | (a)      | (b)      | (c)      | ATE↓         | RPE trans↓   | RPE rot↓     | Abs↓         | δ1.25↑       |
> | ----------------------------- | -------- | -------- | -------- | ------------ | ------------ | ------------ | ------------ | ------------ |
> | Baseline                      |          |          |          | 0.114694     | 0.032125     | 0.347920     | 0.216433     | 0.725167     |
> | Exp-1                         | &#10004; |          |          | 0.114065     | 0.032250     | 0.335198     | 0.215058     | 0.726943     |
> | Exp-2                         |          | &#10004; |          | 0.110530      | 0.033122     | 0.334005     | 0.210339     | 0.722999     |
> | Exp-3                         |          |          | &#10004; | 0.114694     | 0.032125     | 0.347920     | 0.214282     | 0.724084     |
> | Exp-4                         | &#10004; | &#10004; |          | 0.108459     | 0.028906     | 0.281979     | 0.205892     | 0.727616     |
> | Exp-5                         | &#10004; |          | &#10004; | 0.114065     | 0.032250     | 0.335198     | 0.214143     | 0.725534     |
> | Exp-6                         |          | &#10004; | &#10004; | 0.110530     | 0.033122     | 0.334005     | 0.207329     | 0.725784     |
> | DynamicGen (Ours)             | &#10004; | &#10004; | &#10004; | **0.108459** | **0.028906** | **0.281979** | **0.204574** | **0.728961** |
>
> **B. Semantic Novelty**
>
> Our pipeline introduces a sophisticated, multi-level captioning system that moves far beyond the sparse object labels of prior work. This is achieved through several novel techniques:`(a) Semantic-Aware Keyframe Extraction (SAKFE)`, `(b) Hierarchical Prompting (HP)`, `(c) Automated Caption Rephrasing` and `(d) Chain-of-Thought (CoT) Prompting`.
>
> | Evaluation Criteria [3] | **Accuracy**↑ | Completeness↑ | Conciseness↑ | Relevance↑ | Average↑  |
> | ----------------------- | ------------- | ------------- | ------------ | ---------- | --------- |
> | Direct Output           | 79.28         | 76.65         | 73.23        | 80.33      | 77.37     |
> | + (a) SAKFE             | 80.23         | 77.46         | 74.01        | 81.45      | 78.29     |
> | + (b) HP                | 82.57         | 81.42         | 71.17        | 82.56      | 79.43     |
> | + (c) Rephrasing        | 82.48         | 80.50         | 71.86        | 83.27      | 79.53     |
> | + (d) COT               | **84.38**     | **82.09**     | **75.87**    | **85.56**  | **81.97** |
>
> In summary, by demonstrating significant improvements on both a novel cross-modal task and an established fine-tuning task, and by detailing the novel components of our pipeline, we have provided concrete new evidence for both the data contribution of DynamicVerse and the technical contribution of DynamicGen. We thank the reviewer again for prompting this clarification.
>
>
>
> > **Q4: More Dataset Statistics.**
>
> We have performed a detailed characterization of the dynamic content within a significant subset of 1,268 videos (filtered from the original 3,471 in YouTube-VIS). We will include the full statistics in our appendix, and a summary of our findings on the distribution of moving objects and actions is provided below.
>
> (1) Moving Object Distribution: This subset contains 2,158 unique moving object instances across 355 distinct categories. The distribution features a high frequency of common objects followed by a long tail of more specific categories, ensuring a mix of both common and rare examples. The top 5 most frequent categories are: person (16.2%), panda (4.4%), rabbit (2.6%), bird (2.5%), and giraffe (2.2%), showcasing a rich variety of both human and animal subjects.
>
> (2) Action and Motion Distribution: To characterize the "body actions," we analyzed the verbs from our generated captions. The distribution highlights a strong focus on dynamic interactions and object manipulation. The top 5 most frequent actions are: moving (276 instances), holding (178), wearing (144), walking (80), and standing (78). The full distribution covers a wide array of fine-grained actions.
>
> ------
> Reference:
>
> [1] 4D LangSplat: 4D Language Gaussian Splatting via Multimodal Large Language Models. CVPR 2025.
>
> [2] MonST3R: A Simple Approach for Estimating Geometry in the Presence of Motion. ICLR 2025.
>
> [3] G-VEval: A Versatile Metric for Evaluating Image and Video Captions Using GPT-4o. AAAI 2025.

---

> > ### Author Response · Authors · 2025-08-02
> > **A Guide to Our Visual Evidence and Demo Video**
> >
> > Dear Reviewer 1YD4,
> >
> > Thanks again for your constructive review and for raising the important point about visualization. As a well-intentioned reminder, we **did include a demo video** in our supplementary material (suppl. - webpage/index.html), which might have been overlooked. To help fully address your concerns on `Q1: Lack of Video Results / Insufficient Visualization / Robustness in Challenging Scenarios`, we would like to elaborate on what this video and the accompanying figures demonstrate:
> >
> > - **Demo Video (suppl. - webpage/index.html)** showcases the reconstructed moving objects, dynamic scenes, and metric-scale camera trajectories. Specifically, these demos include challenging cases such as significant occlusion (e.g., the "Public Bus" scene) and fast motion (e.g., the "Public Bus" and "Walking Woman" scenes). This video evidence directly demonstrates the high quality of our dataset annotations in these very challenging scenarios.
> > - **Figures 2-4 (suppl.)** showcase our metric-scale 4D reconstruction results on several challenging in-the-wild videos. These qualitative examples highlight our method's ability to produce cleaner static backgrounds and more coherent shapes for dynamic objects, with significantly fewer artifacts like "ghosting" or "streaking". These results serve as compelling evidence that our full pipeline is robust and produces high-fidelity geometric annotations from complex, unconstrained video inputs.
> > - **Figure 5 (suppl.)** provides a direct comparison of our dynamic object segmentation against other methods. We urge the reviewer to note how our approach maintains sharper boundaries and greater temporal consistency, especially for fast-moving objects (e.g., "Sprinting" and "Car Racing Competition" scenes). It demonstrates superior robustness to partial occlusions (e.g., "Figure skating" scene), correctly segmenting partially hidden objects. This precise segmentation is the cornerstone of robust downstream 4D reconstruction.
> >
> > Together, this qualitative evidence visually corroborates the strong quantitative results, confirming that our `DynamicGen` pipeline produces the high-fidelity geometric and semantic annotations that constitute our `DynamicVerse` dataset, proving its suitability for demanding downstream tasks.
> >
> > Best,
> >
> > The DynamicVerse Authors.

---

> > > ### Author Response · Authors · 2025-08-05
> > > **Follow-up regarding our rebuttal for DynamicVerse**
> > >
> > > Dear Reviewer 1YD4,
> > >
> > > We hope this message finds you well. Thank you again for your detailed review and constructive feedback.
> > >
> > > As the discussion period is ending soon, we wanted to gently follow up on our rebuttal and ensure we have fully addressed your concerns. We are particularly keen to clarify the points raised in your "Weaknesses" section.
> > >
> > > 1. **Demo Video and Visualizations (Weakness 1 & 2):** Your primary concern was the lack of a demo video to assess quality and robustness. We wanted to respectfully highlight that a comprehensive demo video **was included in our original supplementary material** at `suppl/webpage/index.html`. This video directly addresses your points by showcasing challenging cases with significant occlusion and fast motion. We believe this visual evidence might have been overlooked and hope it clarifies your concerns about the realism and robustness of our pipeline.
> > > 2. **New Experiments and Clarifications (Weakness 3 & 4):** In response to your other excellent suggestions, our rebuttal also provides:
> > >    - **New cross-modality experiments** to validate our dataset's contribution on an under-explored task, as you suggested.
> > >    - **Detailed ablation studies** to quantify the impact of our novel pipeline components.
> > >    - The specific **loss weight settings** for Eq. (1) and expanded dataset statistics.
> > >
> > > Your feedback has been invaluable. We would be very grateful if you had a moment to consider our clarifications and the new results provided in the rebuttal. We are eager to hear if these updates have addressed your concerns.
> > >
> > > Thank you for your time and consideration.
> > >
> > > Best regards,
> > >
> > > DynamicVerse's authors.

---

> > > > ### Comment · Reviewer_1YD4 · 2025-08-06
> > > >
> > > > Thanks to the authors for the detailed responses to each point. These responses have eliminated my main concerns. I would like to raise my score.

---

> > > > > ### Author Response · Authors · 2025-08-07
> > > > >
> > > > > Thank you for reevaluating our paper and for the improved score. We are grateful for your constructive feedback and will include the suggested experiments in the final version of the manuscript. We appreciate you taking the time to provide such valuable comments!

---

### Official Review · Reviewer_6prH · 2025-06-30

**Clarity:** 4
**Significance:** 3
**Originality:** 3
**Rating:** 5
**Confidence:** 4

**Summary:**

This paper presents DynamicVerse, a new large-scale 4D dataset derived from real-world internet videos, and DynamicGen, the automated pipeline built to create it. DynamicGen annotates videos with a rich set of multi-modal data, including metric-scale geometry, consistent camera poses, and dense, hierarchical text descriptions of scene dynamics. The authors validate the high quality of their dataset by using it to train models that achieve state-of-the-art performance on several downstream benchmarks, such as monocular depth and pose estimation.

**Questions:**

See the weakness section.

**Ethical Concerns:**

["NO or VERY MINOR ethics concerns only"]

**Final Justification:**

I have read the authors' rebuttal and confirm my high rating. The response was exemplary. The authors not only provided a strong justification for the method's generalizability but also went beyond expectations by delivering a full statistical analysis of the dataset and a detailed breakdown of computational costs. These additions significantly improve the paper's transparency and utility, addressing all potential issues. The authors' professionalism and diligence give me full confidence in the final quality of this work.

**Limitations:**

yes

**Paper Formatting Concerns:**

The paper appears to be well-formatted and I did not notice any major violations of the NeurIPS 2025 formatting instructions.

**Quality:**

3

**Strengths And Weaknesses:**

###  **Strengths**

* **Addresses a Significant Problem:** The paper tackles a well-known and critical bottleneck in 4D vision and robotics: the scarcity of large-scale, diverse, and richly-annotated 4D data from real-world scenes. The creation of the DynamicVerse dataset is a valuable and high-impact contribution to the community.

* **Thorough Experimental Validation:** The authors  rigorously validate the quality of their annotation pipeline. By demonstrating state-of-the-art performance on three separate downstream tasks (depth, pose, and intrinsics estimation), they provide strong, quantitative evidence that the generated data is accurate and useful.

* **Novelty in System Integration and Optimization:**  While many components are off-the-shelf, the way they are integrated is novel and non-trivial.

* **Well-Written and Clearly Presented:** The paper is exceptionally well-written and easy to follow. It clearly motivates the problem, logically explains the complex pipeline, and provides detailed appendices that add transparency to the work.

### **Weaknesses**

The paper's primary contributions are strong, but its claims of generalizability and scalability could be better substantiated by addressing the following key points:

* **Generalizability Beyond "In-the-Wild" Internet Videos:** The dataset is sourced from internet videos, which, while diverse, have their own characteristic styles (e.g., camera motions, subjects, video quality). It's worth questioning how well the statistics of this dataset map to other critical domains, such as egocentric robotics or professional film, and consequently, how well models trained on it will generalize.

* **Insufficient Dataset Characterization:** The paper lacks a deep statistical analysis of the DynamicVerse dataset itself, such as detailed distributions of scene/motion types,  clear documentation on data formats and accessibility tools, and a discussion on ethical considerations and potential data source biases.

* **Unreported Inference Speed and Computational Cost:** For a data generation pipeline whose main claim is scalability, the paper provides no analysis of its end-to-end inference time or computational requirements. Understanding the cost to process a single video is crucial for assessing its practical viability and scalability.

---

> ### Author Rebuttal · Authors · 2025-07-31
>
> We sincerely thank the reviewer for their positive and encouraging feedback. We are delighted that the reviewer found our paper to be a high-impact contribution with thorough experimental validation and a clear presentation. We address the remaining points below.
>
> ------
>
> > **Q1: Questions on Generalizability Beyond Internet Videos.**
>
> We thank the reviewer for this insightful question regarding generalizability. The robustness of our pipeline on challenging, in-the-wild data is a key prerequisite for its applicability to demanding domains like egocentric robotics and professional film. We demonstrate how our pipeline's qualitative superiority on core challenges translates to value in these areas.
>
> Our pipeline's ability to achieve more robust moving object segmentation and global refinement directly translates to higher-quality results on complex scenes. The full extent of this robustness is best appreciated through video visualizations. While `the NeurIPS policy prohibits providing images, videos, or other rich media during the rebuttal period`, we are fully committed to significantly expanding our supplementary material.
>
> - **Demo Video (suppl. - webpage/index.html):** This showcases our full 4D reconstruction. Specifically, these demos include challenging cases such as significant occlusion (e.g., the "Public Bus" scene) and fast motion (e.g., the "Public Bus" and "Walking Woman" scenes). Handling such dynamic events is a fundamental challenge in both robotic perception, where tracking objects in cluttered environments is key, and in analyzing cinematic action sequences in film.
> - **Figures 2-4 (suppl.):** These figures highlight our ability to produce clean static backgrounds and coherent shapes for dynamic objects, with minimal "ghosting" or "streaking." This level of fidelity is critical for applications ranging from creating accurate environment models for robotic navigation to generating clean plates for VFX and compositing in film.
> - **Figure 5 (suppl.):** This figure demonstrates our superior dynamic object segmentation, showing sharper boundaries and greater temporal consistency. Such precise, temporally-stable segmentation is vital for reliable object tracking in robotics and for high-quality rotoscoping and character interaction in film production.
>
> In summary, by demonstrating robustness on these core 4D challenges, we provide strong evidence that our `DynamicGen` pipeline and the resulting `DynamicVerse` dataset offer a powerful foundation for both general-purpose 4D understanding and specialized applications in robotics and film.
>
>
>
> > **Q2: Insufficient Dataset Characterization and Documentation**.
>
> We sincerely thank the reviewer for these important suggestions on improving the dataset's characterization and documentation. We agree that providing these details is essential for community adoption and responsible research. We have prepared the following analyses and statements, which will be added to the appendix and our project website in the final version.
>
> **1. Detailed Dataset Characterization**
>
> To provide a concrete statistical analysis as requested, we have characterized a significant subset of our dataset: the 1,268 high-quality videos sourced and filtered from the original 3,471 videos in the YouTube-VIS dataset. Our analysis reveals a rich diversity of objects and actions:
>
> - **Object Distribution:** This subset contains 2,158 unique moving object instances across 355 distinct categories. The distribution is characterized by a high frequency of common objects followed by a long tail of more specific categories, ensuring a mix of common and rare instances. The top 5 most frequent categories are: person (16.2%), panda (4.4%), rabbit (2.6%), bird (2.5%), and giraffe (2.2%). This demonstrates a rich variety of both human and animal subjects.
> - **Action & Motion Distribution:** To characterize the "motion types," we analyzed the verbs from our generated captions. The distribution highlights a strong focus on dynamic interactions. The top 5 most frequent actions are: moving (276 instances), holding (178), wearing (144), walking (80), and standing (78). The full distribution covers a wide array of fine-grained actions.
>
> **2. Data Format and Accessibility Tools**: To ensure our dataset is transparent and easy to use, we will provide comprehensive documentation detailing our data structure. Each scene will be organized in a self-contained directory with annotations saved in standard, easy-to-parse formats: all camera parameters (intrinsics and poses) will be stored in a single `anno.npz` file for efficient loading; per-frame depth maps will be saved as high-precision 16-bit `.png` files; object segmentation masks will be provided as a sequence of `.png` images; and all hierarchical text captions will be organized in a `captions.json` file. To further improve accessibility, we will release a suite of Python scripts for easy data parsing and visualization. Furthermore, the complete `DynamicVerse` dataset will be publicly hosted on the Hugging Face Hub to ensure broad access for the research community.
>
> **3. Ethical Considerations and Data Bias**: We have also drafted a detailed datasheet and ethics statement that will accompany the dataset release. All videos are sourced from public platforms under permissive, non-commercial licenses (e.g., Creative Commons). We acknowledge that any dataset sourced from the internet may reflect the geographical and cultural biases of its source platforms, and we will encourage users to be aware of this. `DynamicVerse` is intended solely for academic and research purposes and not for surveillance or any application that may infringe on personal privacy.
>
> We are confident that these additions will make `DynamicVerse` a more transparent, accessible, and responsible resource for the research community.
>
>
>
> > **Q3: Unreported Inference Speed and Computational Cost**.
>
> We thank the reviewer for highlighting this important point. A clear analysis of the computational cost, including both processing time and memory requirements, is indeed crucial for assessing the practical scalability of our `DynamicGen` pipeline.
>
> To provide concrete and reproducible figures, we processed all 23 videos from the Sintel training set on a single NVIDIA H20 GPU. The table below presents the average processing time and peak VRAM usage for each major module in our pipeline.
>
> | Module                                   | Hardware Used            | Avg. Time / Sintel Video (mins) | Peak VRAM (GB) | Notes                                                |
> | ---------------------------------------- | ------------------------ | ------------------------------- | -------------- | ---------------------------------------------------- |
> | 1. Motion-aware Keyframe Extraction      | 1x H20 GPU               | ~0.1                            | ~10            | Selects representative frames                        |
> | 2. VLM-Based Semantic Analysis (Qwen-VL) | 2x H20 GPU               | ~1.6                            | ~60            | Identifies dynamic elements                          |
> | 3. Moving Object Segmentation (SA2VA)    | 1x H20 GPU               | ~0.8                            | ~30            | Per-object video segmentation                        |
> | 4. Dynamic Bundle Adjustment             | 1x CPU Core + 1x H20 GPU | ~12.2                           | ~30            | Main time bottleneck                                 |
> | 5. Moving Object Captioning              | 2x H20 GPU               | ~2.0                            | ~24            | Object-level descriptions                            |
> | 6. Dynamic Scene Captioning              | 2x H20 GPU               | ~3.0                            | ~40            | Scene-level descriptions                             |
> | 7. Camera Motion Captioning              | 2x H20 GPU               | ~2.0                            | ~40            | Camera-level descriptions                            |
> | 8. Caption Rephrasing                    | 1x H20 GPU               | ~2.0                            | ~24            | LLM-based refinement for consistency and conciseness |
> | **Total (per video)**                    | H20 GPU                  | **~23.7**                       | **~60**        | Peak VRAM, not sum                                   |
>
> ------
>
> We believe this detailed, per-module cost analysis provides the necessary information for the community to assess the practical viability and scalability of our pipeline. We will add this breakdown to our supplementary material for full transparency.

---

> > ### Comment · Reviewer_6prH · 2025-08-05
> >
> > Thanks for the authors' response. It has addressed the majority of my concerns. I will maintain my original score.

---

> > > ### Author Response · Authors · 2025-08-07
> > >
> > > We sincerely appreciate you taking the time to review our work again and for your valuable feedback. Your comments have been very helpful, and we will be sure to address them in our final manuscript. Thank you for your continued high score!

---

> ### Author Response · Authors · 2025-08-04
> **Author Final Reply: Thank You for Championing Acceptance**
>
> Dear Reviewer 6prH,
>
> Thank you for your very positive and encouraging review, and for championing our paper for acceptance.
>
> We are particularly delighted that you found our work to be a **"valuable and high-impact contribution"** that **"addresses a significant problem"** in the community. We also truly appreciate your kind words on our **"thorough experimental validation"** and the paper's **"clear presentation."**
>
> Thank you again for your thoughtful and supportive assessment.
>
> DynamicVerse's authors.

---

### Official Review · Reviewer_6NBV · 2025-07-02

**Clarity:** 3
**Significance:** 2
**Originality:** 3
**Rating:** 4
**Confidence:** 4

**Summary:**

To address the challenge of limited 4D multi-modal data, the paper introduces an automated pipeline that generates 4D annotations and corresponding textual descriptions for input videos by leveraging large vision models, optimization techniques, VLMs, and LMMs. Using this pipeline, the authors compile a large-scale video dataset with detailed annotations, including 4D infos and texture captions. To demonstrate the effectiveness of their approach, they conduct experiments on video depth estimation and camera parameter prediction, showcasing the advantages of their pipeline.

**Questions:**

1. It appears that the authors use VGGT to extract geometric features. Why not compare with more advanced VGGT variants in the camera parameter estimation task?
2. How can the usefulness of the created dataset be demonstrated?
3. What's the core difference compared to prior work Uni4D?
4. How can it be ensured that the video filtering process does not affect the distribution of real-world videos?

**Ethical Concerns:**

["NO or VERY MINOR ethics concerns only"]

**Final Justification:**

Thanks to the authors for the detailed responses to each point. The additional experiments on downstream tasks and the effective use of data from the constructed dataset help better demonstrate its usefulness. The detailed semantic annotations and improved geometry reconstruction further strengthen the dataset. I believe this dataset can benefit the community across a range of downstream tasks. I would like to raise my score.

**Limitations:**

yes

**Quality:**

3

**Strengths And Weaknesses:**

Strengths:
1. The motivation to leverage existing foundation models to overcome the limitations of available 4D data is reasonable
2. The overall pipeline is well designed, effectively leveraging foundation models to obtain initialization, decompose static and dynamic components during bundle adjustment, and incorporate global optimization for refinement. The engineering effort invested in integrating various models into a unified framework is also commendable, as it demonstrates the potential performance gains achievable by combining foundation models with traditional optimization-based methods.
3. The experiments effectively demonstrate the versatility and effectiveness of the proposed pipeline across different tasks including depth estimation and camera parameters prediction

Weaknesses:
1. In the current paper, the goal is to leverage the pipeline to create a large-scale 4D dataset. However, there is no evidence provided to demonstrate that the resulting dataset is actually useful for training models, or that it introduces more challenging data patterns when used for evaluation. The paper does not show whether the annotations generated by the pipeline are of sufficient quality for downstream tasks. Furthermore, it remains unclear whether models trained on this dataset might easily collapse or overfit to certain error patterns.
2. The proposed pipeline for geometry and correspondence is in fact quite similar to the prior work Uni4D, and the quantitative results show that their performances are also comparable. This somewhat diminishes the technical contribution of the pipeline component.
3.It's unclear if the video filtering process ensures that the retained videos are not overly simplified, which could reduce their usefulness for dataset creation.

---

> ### Author Rebuttal · Authors · 2025-07-31
>
> Thank you for your valuable comments and kind words to our work. Below we address specific questions.
>
> ------
> > **Q1: Lack of DynamicVerse Dataset Validation.**
>
> We thank the reviewer for this critical feedback. To directly demonstrate our dataset's utility, we conducted new experiments which provide strong evidence for the quality and effectiveness of the geometric and semantic annotations in DynamicVerse.
>
> **1. Geometric Validation: Fine-tuning MonST3R**：
>
> To validate the quality and utility of our geometric annotations (i.e., metric-scale point maps and camera parameters), we used a subset of 1000 scenes from DynamicVerse to fine-tune the prominent model, MonST3R [1]. As shown in the table below, fine-tuning on our data leads to a performance improvement on the Sintel benchmark. This directly confirms that the geometry in DynamicVerse is accurate and beneficial for improving existing models.
>
> | Method                            | ATE↓  | RPE trans↓ | RPE rot↓ | Abs↓  | δ1.25↑ |
> | --------------------------------- | ----- | ---------- | -------- | ----- | ------ |
> | MonST3R (Original)                | 0.108 | 0.043      | 0.729    | 0.358 | 52.1   |
> | MonST3R + sub-DynamicVerse (Ours) | 0.108 | 0.042      | 0.727    | 0.335 | 58.9   |
>
> **2. Semantic Validation: Downstream Tasks and Direct Evaluation**：
>
> We performed three distinct experiments to validate the high quality of our hierarchical semantic annotations:
>
> **(a) Object-Level Semantics via 4D-LangSplat [2]:** To demonstrate the effectiveness of our dynamic object annotations, we used them to train a 4D-LangSplat model. Our precise object masks and labels enabled the model to achieve superior results in language-grounded 4D scene understanding. This validates that our object-level annotations are highly effective for complex, multi-modal downstream tasks.
>
> | Metrics                                       | Acc(%) ↑ | vIoU(%) ↑ |
> | --------------------------------------------- | :------: | --------- |
> | w/ Original Semantic Annotations for Training |  53.84   | 27.55     |
> | w/ DynamicVerse Semantic Annotations (Ours)   |  64.42   | 51.65     |
>
> **(b) Scene-Level Semantics via G-VEval [3]:** We used the established G-VEval benchmark [3] to quantitatively assess our generated scene-level captions. On a random sample of 100 videos from filtered SA-V data [4], our captions achieved a high score, confirming their coherence, accuracy, and richness.
>
> | Evaluation Criteria   | **Accuracy**↑ | Completeness↑ | Conciseness↑ | Relevance↑ | Average↑ |
> | --------------------- | ------------- | ------------- | ------------ | ---------- | -------- |
> | DynamicVerse Captions | 84.38         | 82.09         | 75.87        | 85.56      | 81.97    |
>
>
> The strong performance across these metrics confirms that our captions are not only factually accurate and relevant to the video content, but also complete in their coverage of events and efficiently concise. This robust, multi-faceted quality makes them highly suitable and reliable for demanding downstream applications.
>
> **(c) Camera-Level Semantics via Human Study:** We conducted a formal human study on 88 videos from our DAVIS subset to quantitatively analyze our camera motion captions. Following the criteria of Clearness, Conciseness, and Grammar & Fluency from prior work [5], the results were excellent: over **60.22%** of captions were rated as both clear and fluent while also receiving high scores for conciseness, confirming the effectiveness of our generation and quality control process.
>
> | Human Evaluation | Rated as Clear | Rated as Fluent | Rated as Concise |
> | ---------------- | :------------: | :-------------: | :--------------: |
> | Camera Captions  |     85.22%     |     89.77%      |      67.04%      |
>
>
>
>
> > **Q2: Limited Novelty, High Similarity to Uni4D.**
>
> We thank the reviewer for the comparison to Uni4D. While both works tackle 4D reconstruction, our motivation and scope are fundamentally different. Uni4D's contribution is a geometric reconstruction framework; our goal is to build a scalable pipeline for a rich, multi-modal dataset with comprehensive annotations (geometry, masks, captions, etc.). This required significant new developments.
>
> **1. Core Geometric Contributions Beyond Uni4D** Our BA pipeline introduces three novel components absent in Uni4D for more robust reconstruction:
>
> - `(a) Epi-Mask-Based Dynamics Filtering`: For cleaner static/dynamic separation before BA, leading to more stable pose estimation.
> - `(b) VLM-Based Semantic Dynamics Analysis`: Uses a VLM for semantic motion understanding, enabling smarter keyframe extraction and more robust object masks.
> - `(c) Optical Flow-Based Sliding Window Global Refinement`: A sliding-window global refinement to enforce temporal consistency and correct drift.
>
> The following ablation study quantifies the significant, measurable impact of each component.
>
> | Components Ablation on Sintel | (a)      | (b)      | (c)      | ATE↓         | RPE trans↓   | RPE rot↓     | Abs↓         | δ1.25↑       |
> | ----------------------------- | -------- | -------- | -------- | ------------ | ------------ | ------------ | ------------ | ------------ |
> | Baseline                      |          |          |          | 0.114694     | 0.032125     | 0.347920     | 0.216433     | 0.725167     |
> | Ablation-1                    | &#10004; |          |          | 0.114065     | 0.032250     | 0.335198     | 0.215058     | 0.726943     |
> | Ablation-2                    |          | &#10004; |          | 0.11053      | 0.033122     | 0.334005     | 0.210339     | 0.722999     |
> | Ablation-3                    |          |          | &#10004; | 0.114694     | 0.032125     | 0.347920     | 0.214282     | 0.724084     |
> | Ablation-4                    | &#10004; | &#10004; |          | 0.108459     | 0.028906     | 0.281979     | 0.205892     | 0.727616     |
> | Ablation-5                    | &#10004; |          | &#10004; | 0.114065     | 0.032250     | 0.335198     | 0.214143     | 0.725534     |
> | Ablation-6                    |          | &#10004; | &#10004; | 0.110530     | 0.033122     | 0.334005     | 0.207329     | 0.725784     |
> | DynamicGen (Ours)             | &#10004; | &#10004; | &#10004; | **0.108459** | **0.028906** | **0.281979** | **0.204574** | **0.728961** |
>
> **2. A Major Contribution Beyond Geometry: Rich Semantic Annotation** Beyond geometry, our primary novelty is a sophisticated semantic annotation pipeline that sets us far apart from Uni4D. Unlike Uni4D's sparse labels, we generate three levels of rich, hierarchical captions (for objects, scenes, and camera motion). This is achieved with novel techniques like `(a) Semantic-Aware Keyframe Extraction (SAKFE)`, `(b) Hierarchical Prompting (HP)`, `(c) Caption Rephrasing` and `(d) Chain-of-Thought (CoT) Prompting`. Our ablation study below validates the effectiveness of each technique in improving the final caption quality.
>
> | Evaluation Criteria | Accuracy↑ | Completeness↑ | Conciseness↑ | Relevance↑ | Average↑  |
> | ------------------- | --------- | ------------- | ------------ | ---------- | --------- |
> | Direct Output       | 79.28     | 76.65         | 73.23        | 80.33      | 77.37     |
> | + (a) SAKFE         | 80.23     | 77.46         | 74.01        | 81.45      | 78.29     |
> | + (b) HP            | 82.57     | 81.42         | 71.17        | 82.56      | 79.43     |
> | + (c) Rephrasing    | 82.48     | 80.50         | 71.86        | 83.27      | 79.53     |
> | + (d) COT           | **84.38** | **82.09**     | **75.87**    | **85.56**  | **81.97** |
>
> In summary, these validated contributions in both geometry and semantics result in a more robust and powerful pipeline for creating rich 4D multi-modal data, representing a significant step forward from prior work.
>
>
>
> > **Q3: Potentially Oversimplified Data due to Filtering.**
>
> Thank you for this question. Our filtering process is a necessary quality control step, not a means of simplification. Our primary goal with filtering is to ensure annotation quality by removing videos that are intractable for current SOTA foundation models (e.g., those with chaotic non-rigid motion or extreme motion blur). This prevents noisy, unusable annotations from contaminating the dataset.
>
> Crucially, our quantitative analysis in the paper (Table 1) empirically proves that DynamicVerse retains the highest degree of dynamism compared to existing datasets, directly addressing the concern of oversimplification. Furthermore, our pipeline's modularity is a key strength; as better foundation models emerge in the future, researchers can seamlessly integrate them into our pipeline to process the very types of complex videos that are currently intractable, making our framework future-proof.
>
>
>
> > **Q4: Choice of Feature Extractor.**
>
> We thank the reviewer for the question and clarify that there is a misunderstanding. VGGT is not used for our final, high-fidelity camera parameter estimation.
>
> Its role is strictly limited to an early, fast pre-filtering step to get a rough initial estimate of intrinsics. For the final, accurate results, we use our sophisticated multi-stage dynamic Bundle Adjustment (BA) pipeline. We deliberately do not use models like VGGT for the final, critical step due to their inherent limitations, such as a lack of metric scale and the requirement for input image cropping, which discards valuable information.
>
> ------
> Reference:
>
> [1] MonST3R: A Simple Approach for Estimating Geometry in the Presence of Motion. ICLR 2025.
>
> [2] 4D LangSplat: 4D Language Gaussian Splatting via Multimodal Large Language Models. CVPR 2025.
>
> [3] G-VEval: A Versatile Metric for Evaluating Image and Video Captions Using GPT-4o. AAAI 2025.
>
> [4] SAM 2: Segment Anything in Images and Videos. ICLR 2025.
>
> [5] CameraBench: Towards Understanding Camera Motions in Any Video.  Arxiv 2025.

---

> ### Author Response · Authors · 2025-08-02
> **Further Evidence: Qualitative Validation of Annotation Quality**
>
> Dear Reviewer 6NBV,
>
> Thanks again for your constructive review. To make sure we fully address your concerns on `Q1: Lack of DynamicVerse Dataset Validation`, we wish to supplement our quantitative experiments with the following qualitative evidence. We strongly encourage you to examine the compelling visual proof within our supplementary material:
>
> - **Demo Video (suppl. - webpage/index.html)** showcases the reconstructed moving objects, dynamic scenes, and metric-scale camera trajectories. Specifically, these demos include challenging cases such as significant occlusion (e.g., the "Public Bus" scene) and fast motion (e.g., the "Public Bus" and "Walking Woman" scenes). This video evidence directly demonstrates the high quality of our dataset annotations in these very challenging scenarios.
> - **Figures 2-4 (suppl.)** showcase our metric-scale 4D reconstruction results on several challenging in-the-wild videos. These qualitative examples highlight our method's ability to produce cleaner static backgrounds and more coherent shapes for dynamic objects, with significantly fewer artifacts like "ghosting" or "streaking" compared to baselines. These results serve as compelling evidence that our full pipeline is robust and produces high-fidelity geometric annotations even from complex, unconstrained video inputs.
> - **Figure 5 (suppl.)** provides a direct comparison of our dynamic object segmentation against other methods. We urge the reviewer to note how our approach maintains sharper boundaries and greater temporal consistency, especially for fast-moving objects (e.g., "Sprinting" and "Car Racing Competition" scenes). Crucially, it demonstrates superior robustness to partial occlusions (e.g., "Figure skating" scene), correctly segmenting objects even when parts are momentarily hidden. This precise segmentation is the cornerstone of robust downstream 4D reconstruction.
>
> Together, this qualitative evidence visually corroborates the strong quantitative results, confirming that our `DynamicGen` pipeline produces the high-fidelity geometric and semantic annotations that constitute our `DynamicVerse` dataset, proving its suitability for demanding downstream tasks.
>
> Best,
>
> DynamicVerse's authors.

---

### Note · Authors · 2025-08-15

Dear Reviewers, ACs, and SACs,

We are profoundly grateful for your insightful feedback, which was instrumental in strengthening our paper. We are especially encouraged by the strong consensus that emerged among the reviewers regarding the significance of our work and the quality of its execution. The recognition of our project as a **"valuable and high-impact contribution"** that **"addresses a significant problem"** (`6prH`) is particularly gratifying, as it affirms the core motivation behind our research. Furthermore, we sincerely appreciate the acknowledgment of our technical approach; hearing that our **"solid pipeline"** (`uDiP`) is considered **"well designed"** (`6NBV`) and highlights **"thoughtful engineering"** (`uDiP`) validates the complex integration and novel components that underpin our framework. Ultimately, we are most heartened by the belief that our resulting dataset and tools will **"benefit the community"** (`6NBV`). Fulfilling this goal was our primary objective from the outset, and this validation gives us great confidence in the future utility of our work.

Our rebuttal addressed all concerns with new experiments and clarifications:

**1. Validated Dataset Utility:** We ran new experiments demonstrating `DynamicVerse`'s practical utility. Our geometric annotations were shown to improve a prominent model (MonST3R) on the Sintel benchmark, and our semantic annotations significantly boosted performance on cross-modal tasks like 4D-LangSplat.

**2. Clarified Technical Novelty:** In response to Reviewer `6NBV`'s comparison to Uni4D, we detailed our contributions beyond system integration. Our ablation studies validated the impact of novel components for both geometric reconstruction (e.g., dynamic filtering) and our sophisticated semantic pipeline (e.g., SAKFE), which ensure high-quality results.

**3. Enhanced Transparency:** For reproducibility, and as requested by Reviewers `6prH` and `uDiP`, we provided details on computational costs, dataset statistics, failure cases, and our human-in-the-loop review process. We also thank Reviewer `1YD4` for the engagement that helped resolve robustness concerns via our supplementary video.

The invaluable dialogue converted all initial concerns into strong, unanimous support. We believe these revisions solidify our work as a foundational contribution to **4D multimodal modeling**, and thank you for your thoughtful re-consideration.

Best regards,

The Authors.

---

### Decision · Program_Chairs · 2025-09-17

**Decision:**

Accept (poster)

**Comment:**

The paper addresses the critical issue of the lack of large-scale, in-the-wild 4D datasets and proposes a pipeline which runs foundation models as well as windowed bundle adjustment on in-the-wild videos where dynamic objects are determined through a series of VLM questions. This then produces a large-scale dataset (100k videos) with high quality pseudolabels.

The paper received positive reviews (3 borderline accept, 1 accept) with reviewers commending the paper's task and approach. The main concerns expressed in the review process was with respect to the verification of the quality of the labels. In particular, that there was no experiment which showed that training with this data could improve prior models. The authors answered this in the rebuttal with 4 new experiments. While the results could be stronger, this does address this issue.

I advocate for acceptance, based on the added verification in rebuttal. I would strongly encourage the authors to rewrite the paper to include these because they substantially improve the paper.